# Formation of Nighttime Sulfuric Acid from the Ozonolysis of Alkenes in Beijing

Yishuo Guo[1], Chao Yan[1,2,*], Chang Li[1], Zemin Feng[1], Ying Zhou[1], Zhuohui Lin[1], Lubna Dada[2], Dominik Stolzenburg[2], Rujing Yin[3], Jenni Kontkanen[2], Kaspar R. Daellenbach[2], Juha Kangasluoma[1,2], Lei Yao[2], Biwu Chu[2], Yonghong Wang[2], Runlong Cai[2], Federico Bianchi[2], Yongchun Liu[1] and Markku Kulmala[1,2]

[1] Aerosol and Haze Laboratory, Beijing Advanced Innovation Center for Soft Matter Science and Engineering, Beijing University of Chemical Technology, Beijing, China

[2] Institute for Atmospheric and Earth System Research / Physics, Faculty of Science, University of Helsinki, Finland

[3] State Key Joint Laboratory of Environment Simulation and Pollution Control, State Environmental Protection Key Laboratory of Sources and Control of Air Pollution Complex, School of Environment, Tsinghua University, Beijing, China

*Correspondence to: Chao Yan (chao.yan@helsinki.fi)

**Abstract.** Gaseous sulfuric acid (SA) has received a lot of attention for its crucial role in atmospheric new particle formation (NPF), and for this reason, studies until now have mainly focused on daytime SA when most NPF events occur. While daytime SA production is driven by $SO_2$ oxidation of OH radical from photochemical origin, the formation of SA during night and its potential influence on particle formation remains poorly understood. Here we present evidence for significant nighttime SA production in urban Beijing during winter, yielding concentrations between 1.0 and $3.0 \times 10^6$ $cm^{-3}$. We found a high frequency (~ 30%) of nighttime SA events, which are defined by the appearance of a distinct SA peak observed between 20:00 and 04:00 local time, and with the maximum concentration exceeding $1.0 \times 10^6$ $cm^{-3}$. These events mostly occurred during unpolluted nights with low vapor condensation sink. Furthermore, we found that under very clean conditions (visibility > 16.0 km) with abundant ozone (concentration > $2.0 \times 10^{11}$ $cm^{-3}$, ~ 7 ppb), the overall sink of SA was strongly correlated with the products of $O_3$, alkenes and $SO_2$ concentrations, suggesting that the ozonolysis of alkenes played a major role in nighttime SA formation under such conditions. This is in light with previous studies showing that the ozonolysis of alkenes can form OH radical and stabilized Criegee intermediate (sCI), both of which are able to oxidize $SO_2$ leading to SA formation. However, we also need to point out that there exist additional sources of SA under more polluted condition, which are not investigated in this study. Moreover, there was a strong correlation between SA concentration and the number concentration of sub-3 nm particles in both clean and polluted nights. Different from forest environments, where oxidized biogenic vapors are the main driver of nighttime clustering, our study demonstrates that the formation of nighttime cluster mode particles in urban environments is mainly driven by nighttime SA production.

**Keywords:** nighttime SA, urban environment, ozonolysis of alkenes, sub-3 nm particles

**1. Introduction**

Atmospheric aerosol particles have considerable impact on global climate by directly affecting the radiation balance of the earth and by indirectly acting as cloud condensation nuclei (Stocker et al., 2014). The number concentration of these aerosol particles depends to a large extent on the atmospheric new particle formation (NPF), which includes gas-phase nucleation and subsequent growth of newly formed particles. Studies over the past twenty years have shown that the SA is the major gaseous
precursor of NPF in most environments inside the continental boundary layer (Lee et al., 2019). Sulfuric acid driven NPF can proceed as SA-water binary nucleation, SA-water-ammonia ternary nucleation (Kirkby et al., 2011), SA-amine-water nucleation (Almeida et al., 2013;Kuerten et al., 2014), SA-organics nucleation (Riccobono et al., 2014), and SA-organics-amonia nucleation (Lehtipalo et al., 2018) and $H_2SO_4$-$H_2O$-$NH_3$-amine nucleation (Myllys et al., 2019;Yu et al., 2012). Both the nucleation rate ($J_{nuc}$) and the initial growth rate of newly formed particles tends to have a power-law relationship with the
SA concentration: $J_{nuc} = k \times SA^{\alpha}$, where the activation nucleation is dominant when $\alpha \approx 1$ (Kulmala et al., 2006), the kinetic nucleation is dominant when $\alpha \approx 2$ (Riipinen et al., 2007;Paasonen et al., 2009;Erupe et al., 2010) and the thermodynamic nucleation becomes more crucial when $\alpha$ is larger than 2.5 (Wang et al., 2011).

Due to the importance of SA for NPF, accurate and reliable measurement of SA is of great importance. Up to now, ambient SA concentrations have been reported for many sites (Weber et al., 1997;Weber et al., 1998;Weber et al., 1999;Paasonen et
al., 2010;Jokinen et al., 2018;Fiedler et al., 2005;Eisele et al., 2006;Boy et al., 2008;Iida et al., 2008;Wang et al., 2011;Kuerten et al., 2016;Yao et al., 2018;Mauldin et al., 2001;Erupe et al., 2010;Yu et al., 2014). These studies indicate that the concentration level of SA in the atmosphere is closely related to human activities. In general, daytime SA concentration is around $10^5$ cm$^{-3}$ in pristine Antarctica region (Mauldin et al., 2001), $10^6$ cm$^{-3}$ in remote continental, remote marine and forest regions (which are less affected by human activities) and $10^7$ cm$^{-3}$ in urban and rural agricultural lands (which are influenced
dominantly by human activities, (Birmili et al., 2003;Fiedler et al., 2005;McMurry et al., 2005;Iida et al., 2008;Erupe et al., 2010;Paasonen et al., 2010;Wang et al., 2011;Yao et al., 2018)). SA generally shows a distinct diurnal pattern correlating with radiation (Lu et al., 2019) with typical concentrations between $10^6$ to $10^7$ cm$^{-3}$ during daytime and $10^4$ to $10^6$ cm$^{-3}$ during nighttime. The seasonal variation of SA is only reported in very few studies, showing higher concentrations during spring and summer than in winter and autumn (Erupe et al., 2010).

The strong connection between SA and NPF has led previous studies to mostly focus on understanding the SA formation in the daytime. However, recent observation on the formation of sub-3 nm particles have shown that these cluster mode particles also exist with high concentration during the night (Junninen et al., 2008;Lehtipalo et al., 2011;Kulmala et al., 2013;Kecorius et al., 2015;Mazon et al., 2016;Yu et al., 2014) and sometimes even nighttime particle nucleation events can be clearly distinguished. In boreal forest environments, nighttime cluster formation can be attributed to highly oxygenated organic
molecules (HOMs) (Kammer et al., 2018;Rose et al., 2018). However, the sources of SA and its role in the particle formation

during the nighttime remain largely unresolved, both of which are the focus of this work. In this study, we show noticeable nighttime sulfuric acid sometimes increase of SA during the nighttime in urban Beijing. We further investigate the main sources of SA and demonstrate its role in the nocturnal formation of sub-3nm clusters.

## 2. Nighttime Sulfuric Acid Formation

During the daytime, gaseous SA is primarily a photochemical product generated from the oxidation of $SO_2$ by OH radical, while at nighttime, SA is highly associated with non-photochemical oxidants, most likely the non-photochemical OH radical and stabilized Criegee intermediate (sCI) (Mauldin et al., 2012;Taipale et al., 2014). And that non-photochemical pathway has been shown important in parameterizing the SA formation the boreal forest environment (Dada et al., 2020). The non-photochemical oxidation pathway mainly includes the following five reactions. First, the production of (stabilized) Criegee

Intermediates by the ozonolysis of alkenes:

$$\text{Alkene} + O_3 \xrightarrow{k_1} \varphi sCI + (1 - \varphi)CI + RCHO \quad \text{R1}$$

Then, the direct oxidation of $SO_2$ by long lived sCI:

$$sCI + SO_2 \xrightarrow{k_2} SO_3 + RCHO \quad \text{R2}$$

Or the alternative oxidation of $SO_2$ by OH radicals formed from decomposition of (stabilized) Criegee Intermediates:

$$sCI \xrightarrow{k_3} \cdot OH + R_1COR_2 \quad \text{R3}$$

$$CI \xrightarrow{k_4} \cdot OH + R_1COR_2 \quad \text{R4}$$

$$SO_2 + \cdot OH + O_2 \xrightarrow{k_5} HO_2 \cdot + SO_3 \quad \text{R5}$$

where $k_i$ is the rate constant of each reaction, $\varphi$ is the yield of sCI in the ozonolysis of alkenes, and CI is the chemically activated Criegee intermediate. The lifetime of $SO_3$ due to its fast reaction with $H_2O$ to form SA is approximately $2 \times 10^{-4}$ s

(see detailed calculation in supplement Section S1), which indicates that this reaction is so fast that almost all $SO_3$ will be instantaneously converted to SA. In this case, the oxidation of $SO_2$ is the rate-limiting step in the formation of SA. Currently, only limited types of sCI has been studied: isoprene-derived sCI (Neeb et al., 1997;Zhang et al., 2002;Atkinson et al., 2006;Newland et al., 2015b), monoterpene-derived sCI (Hatakeyama et al., 1984;Rickard et al., 1999;Zhang and Zhang, 2005;Mauldin et al., 2012;Sipila et al., 2014;Vereecken et al., 2017) and the simplest sCI including $CH_2COO$, $CH_3CHOO$ and

$(CH_3)_2COO$ (Hatakeyama et al., 1984;Hasson et al., 2001;Welz et al., 2012;Taatjes et al., 2013;Welz et al., 2014;Newland et al., 2015a;Vereecken et al., 2017). Based on the aforementioned studied, the yield $\varphi$ of sCI can vary from 0.1 to 0.65 and the rate constants for different reactions span over several orders of magnitude, for $k_1$ from $1.6 \times 10^{-18}$ to $2.5 \times 10^{-16}$ $cm^3 s^{-1}$ and for $k_2$ from $1.4 \times 10^{-13}$ to $2.2 \times 10^{-10} cm^3 s^{-1}$. The yield of OH radical from ozonolysis of different type of alkenes also covers a wide range with values of $0.68 - 0.91$, $0.24 - 0.35$, $0.25 - 0.44$, $0.32 - 0.40$ and $0.33 - 1.00$ for α-pinene, β-pinene,

isoprene, propene and other C4-C6 alkenes respectively (Atkinson et al., 1992;Aschmann, 1993;Chew and Atkinson, 1996;Rickard et al., 1999;Siese et al., 2001;Witter et al., 2002;Berndt et al., 2003;Aschmann et al., 2003;Nguyen et al., 2009;Malkin et al., 2010). Moreover, the bimolecular reaction and decomposition reactivity of sCI is highly structure-dependent. sCI with more complicated substituent groups tend to react with $H_2O$ more slowly (Huang et al., 2015), decompose

faster (Fenske et al., 2000;Hasson et al., 2001) and more likely to react with $SO_2$. There were also studies showing that the

reactions between sCI and $SO_2$ were pressure and temperature dependent and were commonly affected by the presence of water and other constituents (Kotzias et al., 1990;Sipila et al., 2014).

## 3. Ambient Observations

The continuous and comprehensive measurements were conducted at the west campus of Beijing University of Chemical Technology (39.95 °N′, 116.31 °E′). Here we investigate the time period from 18[th] January to 16[th] March 2019. The measuring

instruments are located on the fifth floor, which is about 15 m above the ground level. This station is a typical urban site, which is around 130 m to the nearest Zizhuyuan Road, 550 m to the West Third Ring Road, and surrounded by commercial properties and residential dwellings.

### 3.1 Measurement of Sulfuric Acid with CI-APi-TOF

Sulfuric acid is measured by a long time-of-flight chemical ionization mass specter (LTOF-CIMS, Aerodyne Research, Inc.)

equipped with a nitrate chemical ionization source. The basic working principle of this instrument can be found elsewhere (Jokinen et al., 2012). In our measurement, we draw air through a stainless-steel tube with a length of 1.6 m and a diameter of 3/4 inch with a flowrate at 7.2 L min$^{-1}$. In addition, we have implemented a flush plate (Karsa Inc.) to effectively remove water molecules entering the instrument, which is found necessary to maintain a continuous measurement.

The quantification of sulfuric acid is derived from the ratio of bisulfate ions (with counting rates unit in ions·s$^{-1}$) relative to

primary ions as follows:

$$[\text{H}_2\text{SO}_4] = \frac{\text{HSO}_4^- + \text{H}_2\text{SO}_4\text{NO}_3^-}{\text{NO}_3^- + \text{HNO}_3\text{NO}_3^- + (\text{HNO}_3)_2\text{NO}_3^-} \times \text{C}$$

The calibration factor, C, is determined from direct calibration by injecting gaseous sulfuric acid of known amounts into the instrument (Kurten et al., 2012). The diffusional wall loss of the 1.6 sampling line is 0.2423, and after taking into account of it, we a value of $6.07 \times 10^9$ cm$^{-3}$ as the final calibration coefficient.

### 3.2 Measurement of Alkenes with SPI-MS

Six alkenes are analyzed in this study, i.e., including propylene, butylene, butadiene, isoprene, pentene and hexene, which were detected by a single photon ionization time-of-flight mass spectrometer (SPI-MS 3000, Guangzhou Hexin Instrument Co., Ltd., China) (Gao et al., 2013). It should be mentioned that this instrument cannot distinguish conformers, and therefore the pentene and haxene could also be cyclopentane and cyclohexane, respectively. A polydimethylsiloxane (PDMS) membrane

sampling system is used. As the PDMS membrane has better selective adsorption to volatile organic compounds (VOCs), VOC molecules can be concentrated after diffusing and desorbing from the membrane under vacuum sampling condition. In this way, the detection limit of VOCs can be improved. Then the gas molecules are guided to an ionization chamber through a 2 mm-diameter stainless steel capillary, where VOC molecules are ionized by the vacuum ultraviolet (VUV) light with an

ionization energy smaller than 10.8eV. For the detection of positive ions, two microchannel plates (MCPs, Hamamatsu, Japan) assembled with a chevron-type configuration are employed. An analog to digital converter (ADC) was used to measure and record the output current signal from the MCPs.

Alkenes concentrations are quantified by performing a direct calibration. The PAMS (Photochemical Assessment Monitoring Stations) and TO-15 environmental gases (including 57 and 65 types of VOCs separately, Linde Gas North America LLC, USA) are used as two standard gases with ultra-high-purity nitrogen as the carrier gas. Gas with different concentrations of VOC standards is produced by mixing a constant carrier gas with standard gas of varying flow rates. The calibration coefficient is further calculated from the ratio between the actual concentration and the ion intensity.

### 3.3 Other Ancillary Measurements

The number concentration of clusters with the size range of 1.30~2.45 nm was measured with a Particle Sizer Magnifier (PSM) (Vanhanen et al., 2011), and the integrated number concentration of particles from PSM is referred as $N_{Sub-3nm}$ in the following. The number size distributions of aerosol particles from 6 to 840 nm and from 0.52 to 19.81 μm are measured by the Differential Mobility Particle Sizer (DMPS) (Aalto et al., 2001) and the Aerodynamic Particle Sizer (APS) (Armendariz and Leith, 2002) respectively. Meteorological parameters are measured with a weather station (AWS310, Vaisala Inc.) located on the rooftop of the building. These parameters include the ambient temperature, relative humidity (RH), pressure, visibility, UVB radiation, as well as horizontal wind speed and direction. Trace gas concentrations of carbon monoxide (CO), sulfur dioxide ($SO_2$) nitrogen oxides ($NO_x$) and ozone ($O_3$) are monitored using four Thermo Environmental Instruments (models 48i, 43i-TLE, 42i, 49i, respectively). Calibration of these instruments are performed monthly using the standard gases of known concentrations. The mass concentration of $PM_{2.5}$ is directly measured with a Tapered Element Oscillating Microbalance Dichotomous Ambient Particulate Monitor (TEOM 1405-DF, Thermo Fisher Scientific Inc, USA) with a total flow rate of 16.67 L/min. In addition, the loss rate of gas-phase sulfuric acid described by condensation sink (CS) is calculated based on the size distribution data from DMPS and APS (Kulmala et al., 2001).

## 4. Results and Discussions

### 4.1 Definition of Nighttime Sulfuric Acid Event

An overview of our measurements during 18th January to 16th March 2019 is shown in Fig. 1. As our measurement period overlaps with the heating period in Beijing (from 15th November 2018 to 15th March 2019), the $SO_2$ level during the measurement period was higher than that of other periods in one year (from 16th March to 31st May 2019) (Fig. S1 in the Supplement).

In this work, the nighttime window is defined between 20:00 and 04:00 (following day) to exclude any possible influence of photochemistry. Fig. 2 shows the diurnal variation of SA concentration on one typical SA event night (14 March 2019) and one typical SA non-event night (3 February 2019). Overall, nighttime SA concentrations vary between $3.0 \times 10^5$ and $3.0 \times 10^6$

cm⁻³ in our measurement period. The nighttime event in Fig. 2 shows a distinct SA peak at around 22:00 with a maximum SA concentration of around $3.0 \times 10^6$ cm⁻³, which is almost half of the daily maximum value. While in the non-event case, SA continues decreasing throughout the evening, reaching a minimal value of $3.0 \times 10^5$ cm⁻³. A nighttime SA event is defined when both of the following two criteria are both met: (a) there is a distinct peak during the nighttime hours, (b) the SA concentration exceeds $1.0 \times 10^6$ cm⁻³. The nights without distinct peaks are classified as SA non-event nights, and if a peak is identified but it

does not meet criterion (b), the night is classified as an undefined night. Out of all 56 nights studied, there are in total 18 SA event nights, 16 non-event nights, and 22 undefined nights (listed in Table. S1). Thus, the overall frequency of nighttime SA events during our observation period is 32%, which means that nearly a third of nights during our observation period had distinct nighttime sulfuric acid peaks.

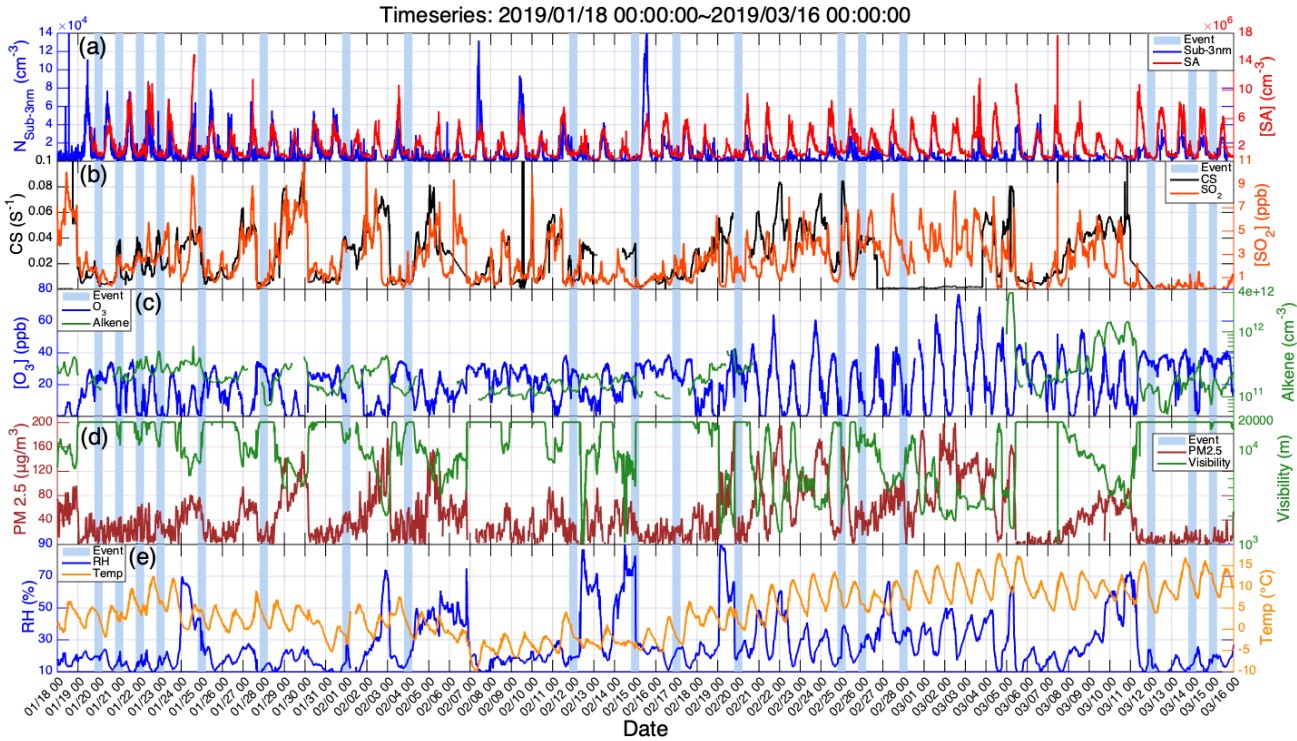

**Fig. 1.** Overview of different parameters measured from 18$^{th}$ January, 2019 to and 16$^{th}$ March, 2019 for (a) SA concentration and particle number concentration of sub-3nm particles ($N_{Sub-3nm}$, measured by PSM), (b) CS and SO$_2$ concentration, (c) concentration of O$_3$ and Alkenes, (d) PM$_{2.5}$ and visibility, and (e) relative humidity (RH) and temperature. The light blue bars represent nights with nighttime SA events.

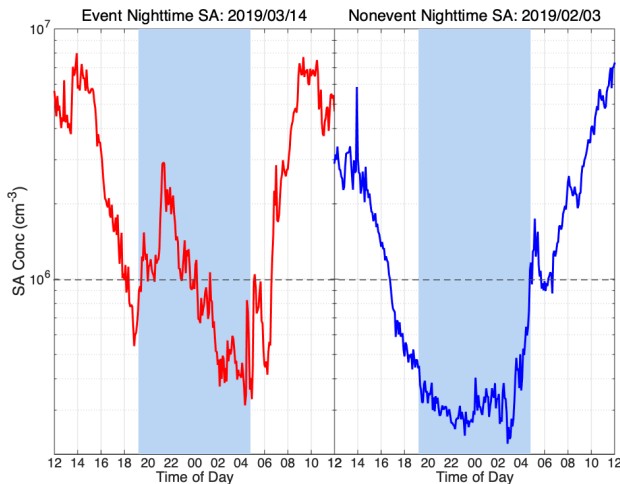

**Fig. 2** Daily variation of SA concentration on a typical night with a nighttime SA event (red line, 14$^{th}$ March, 2019) and on a non-event night

(blue line, 3$^{rd}$ February, 2019). The shaded blue area shows the period that is considered as nighttime in this study.

**4.2 Features of Nighttime Sulfuric Acid Event**

We further analyzed the features of nighttime SA event nights based on the above mentioned 18 event nights and 16 non-event nights (Fig. 3). On SA event nights, the mass concentration of $PM_{2.5}$, the mixing ratio of $NO_x$, CS and RH were clearly lower, and visibility was clearly higher than on non-event nights. This suggests that nighttime SA events tend to occur under clean conditions. In addition, higher concentrations of $O_3$ were associated with SA event nights, whereas the concentration of $SO_2$ did not vary as much between SA event and non-event nights. This indicates that at most of the time, the concentration of $SO_2$ is not the dominant factor that explains the variation of nighttime SA.

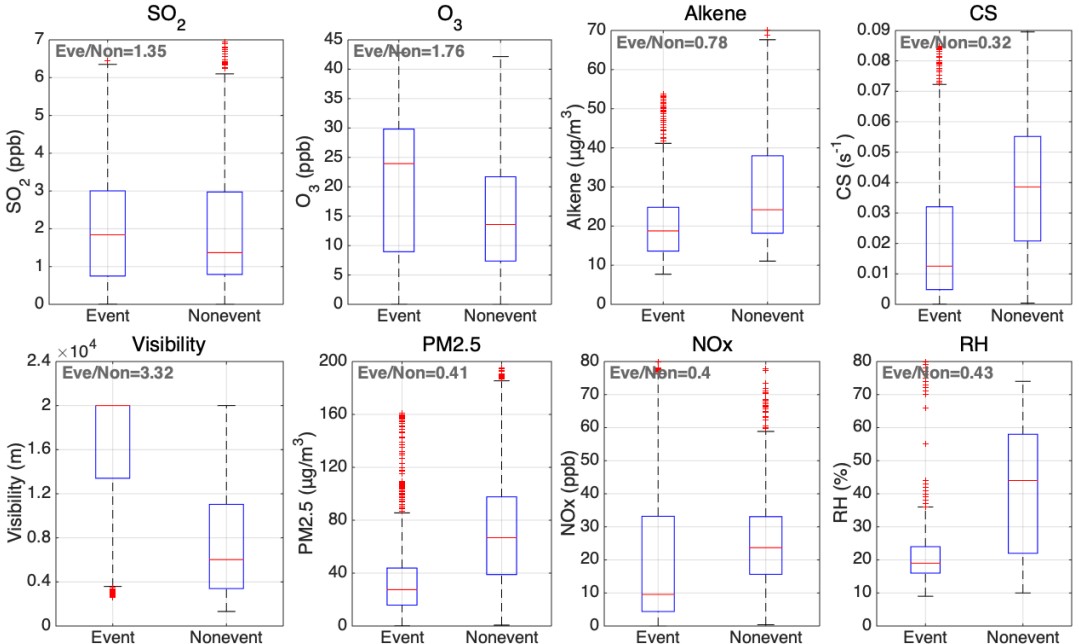

**Fig. 3** Boxplots for the concentrations of $SO_2$, $O_3$ and alkenes, CS, visibility, $PM_{2.5}$, $NO_x$ concentration and RH during nighttime SA event and non-event nights. In each plot, the red line is the median value, the bottom and the top blue lines are the 25 and 75 percentiles, and the whisker range covers the $\pm 2.7\sigma$ of the data. Outliers are the ones out of the $\pm 2.7\sigma$ range of all selected data. The dark gray values on the top left corners are the ratios between median values of event and non-event days.

We further investigated the determining factor for the occurrence of SA events by looking into different variables during the SA event nights. CS measurements were available for 13 event nights, during which 15 SA peaks were observed. In general, we found that eight events (53%) were mainly associated with the decrease of CS. This is demonstrated in Fig. 4, where the nighttime events as well as the simultaneous decrease of CS are highlighted with green dots. Four other cases (27%) were mostly due to the increase of $SO_2$ concentration (Fig. S2), and the remaining three cases were likely synergistically caused by $SO_2$, $O_3$, alkenes, CS and other parameters. More details are provided in Table S2.

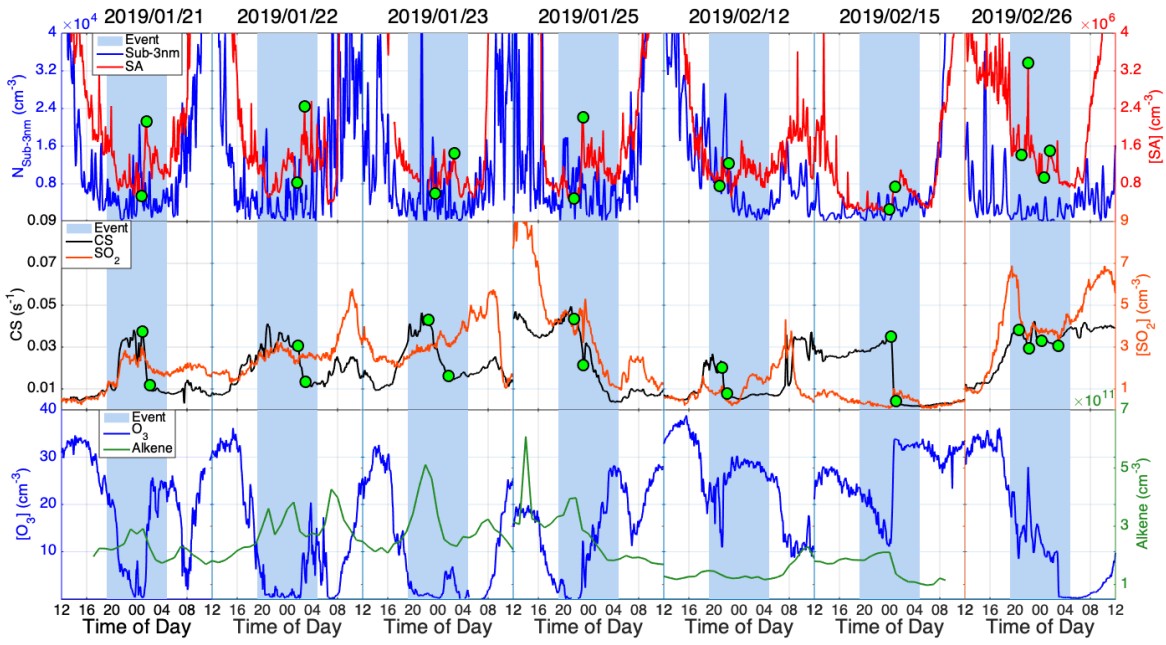

**Fig. 4** Daily time-series of different variables on nighttime SA event days when SA events occurred under CS decrease conditions. The top panel shows the $N_{Sub-3nm}$ and SA concentration, the middle panel shows CS and $SO_2$ concentration, and the bottom panel shows the concentration of $O_3$ and alkenes. Green dots show times when CS started to drop and reached its minimum value.

### 4.3 Source and Sink Balance for Nighttime Sulfuric Acid: Importance of Alkene Ozonolysis

As discussed above, nighttime SA events mainly occurred under clean conditions with low CS values. Therefore, we classified all the nighttime data set into three groups according to the air pollution level, which is assessed by visibility. The division standards for pollution level is explained in detail in Section S3. The clean (named Clean-1), mildly polluted and heavy polluted conditions are defined by visibility values which are larger than 12.0 km, in the range of 4.0 - 12.0 km and smaller than 4.0 km respectively. Accordingly, data points under each condition took up 48%, 25% and 27% of all data points.

After classifying the data set into groups based on the pollution level, the balance between SA source and sink for each group was investigated separately. The ozonolysis of alkenes under dark conditions is capable of generating sCI as well as OH radical, both of which are able to oxidize $SO_2$ to form gaseous SA. However, the yields of both sCI and recycled OH radical remain largely unquantified. Therefore, we do not attempt to distinguish the contribution of sCI and OH radical on SA formation in this study, but rather treat them as a "bulk oxidant" and use an empirical parameter $k_{app}$ to account for both oxidation pathways. Accordingly, the source term (production rate) of SA can be expressed as $k_{app} \cdot [SO_2] \cdot [O_3] \cdot [Alkene]$, where $k_{app}$ is an overall empirical parameter that takes into account the yields of OH radical and sCI as well as the rate constants of their reactions with $SO_2$. The sink term (loss rate) consists of two parts: the condensation of SA onto particles ($[SA] \cdot CS$) and the collision of SA monomers with each other to form SA dimers ($\beta \cdot [SA]^2$). In reality, SA monomer also collides with SA dimers and larger clusters, but due to the low concentration of SA clusters, those collisions are negligible compared to other losses. In a polluted environment where strong stabilizers of SA exist, the formation rate of stable SA dimer is close to the collision limit (Yao et al., 2018). Therefore, $\beta$ can be taken as the hard-sphere collision rate, which is calculated to be $3.46 \times 10^{-10}$ $cm^3$ $s^{-1}$ (Seinfeld and Pandis, 2016). Under pseudo-steady-state (PSS) assumption (see Section S2 for detailed disscussion about PSS assumption), the source-sink balance of SA can be expressed as follows:

$$k_{app} \cdot [\text{Alkene}] \cdot [O_3] \cdot [SO_2] = [SA] \cdot CS + \beta \cdot [SA]^2$$

Fig. 5 shows the nighttime correlation between SA source term and sink term under different pollution levels, with the data binned by SA sources. If the source and the sink term correlate, then the slope represents the overall apparent rate constant ($k_{app}$) concerning the reaction between oxidants (OH radical and sCI) from ozonolysis of alkenes and $SO_2$ for this specific pollution level.

Under Clean-1 conditions (Fig. 5 (a)), the source term and sink term have a good linear correlation ($R^2=0.97$) in the source range of $1.0\times10^{33}$ - $4.5\times10^{33}$ $(cm^{-3})^3$, while the balance is broken up outside of this range. These uncorrelated data points outside of this range appear when the visibility was smaller than 16.0 km (Fig. S4 (a)) along with higher concentrations of $NO_x$ and NO ($[NO_x] > \sim 40$ ppb, Fig. S4 (b) and ($[NO] > \sim 3$ ppb, Fig. S4 (c)). High $NO_x$ levels always relate to pollution, and NO will consume $O_3$, leading to much lower $O_3$ concentration (marked by blue empty circles in Fig. S4 (a)). Thus, we redefined the criterion for clean condition (Clean-2) so that visibility needs to be larger than 16.0 km and $O_3$ concentration higher than $2.0\times10^{11}$ $cm^{-3}$ ($\sim 7$ ppb). These conditions account for 38% of all data. Fig. 5 (b) shows the good correlation between $[SO_2] \cdot [O_3] \cdot [\text{Alkene}]$ source term and $[SA] \cdot CS + \beta \cdot [SA]^2$ sink term ($R^2=0.97$) under the redefined clean condition over the entire source range. This suggests that the ozonolysis of alkenes indeed have a dominant contribution to the formation of SA during nighttime under very clean conditions. Generally, the sink term of SA condensation onto particles took up 95.5% for Clean-2 condition and increase to 99.7% for heavy polluted condition. The fitted value, 95% confidence bounds, uncertainty of $k_{app}$ and correlation coefficient $R^2$ for Clean-1 and Clean-2 condition are listed in Table S3.

If we compare the SA source and sink correlation between the Clean-1 (Fig. 5 (a)) and Clean-2 (Fig. 5 (b)) condition, it is obvious that the slope of the linear region of Clean-1 condition data points ($2.7\times10^{-30}$ $cm^6 s^{-2}$) matches well with the slope of Clean-2 condition data points ($2.6\times10^{-30}$ $cm^6 s^{-2}$), which further confirms the reliability of the balance between $[SO_2] \cdot [O_3] \cdot [\text{Alkene}]$ source term and $[SA] \cdot CS + \beta \cdot [SA]^2$ sink term under clean condition. Then we then took data from another period to further evaluate the reliability of the proposed source and sink balance. For spring period from 20[th] March to 20[th] May 2019 (Fig. S7 (b)), there is also a good linear correlation ($R^2=0.98$) when source term is smaller than $6.0\times10^{33}$ $(cm^{-3})^3$ with $k_{app}$ of $1.5\times10^{-30}$ $cm^6 s^{-2}$. Although the fitted $k_{app}$ values deviate beween these two periods, they are in the same order of magnitude. During spring period, apart from the above-mentioned six alkenes, biogenic emitted monoterpenes, which cannot be measured by our instrument and therefore are not included in the source term, start to have a bigger contribution, which likely leads to the deviation of $k_{app}$. Besides, the yield of sCI and the rate constant between sCI and $SO_2$ are to some extent temperature-dependent (Berndt et al., 2014), which may further explain at least a part of the difference of $k_{app}$ between winter and spring observations.

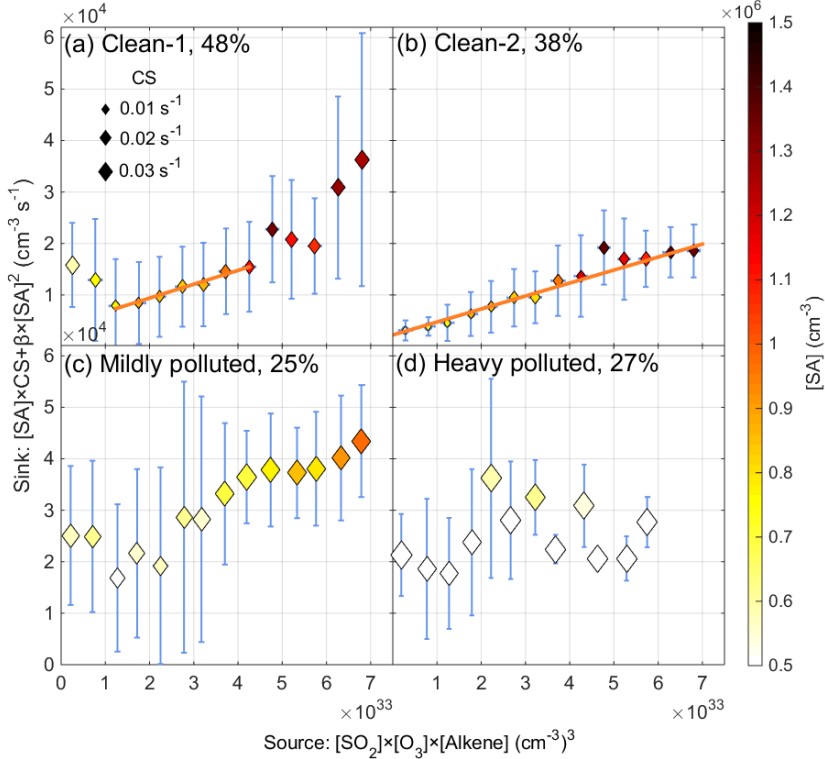

**Fig. 5** Correlation between the source term ($[SO_2]\cdot[O_3]\cdot[Alkene]$) and sink term ($[SA]\cdot CS+\beta\cdot[SA]^2$) of SA under PSS assumption during nighttime (20:00-04:00) from 18th January to 16th March, 2019 for (a) Clean-1 condition, (b) Clean-2 condition, (c) mildly polluted condition and (d) heavy polluted condition. Note that the data points are mean values of corresponding bin ranges instead of the original, high time resolution data (Fig. S5). The error bars are the standard deviation of all data points in each bin.

Under mildly polluted conditions (Fig 5 (c)), the source and sink term also have a good linear correlation when source value exceeds $2.5\times10^{33}$ (cm$^{-3}$)$^3$, while under heavy polluted conditions, the $[SO_2]\cdot[O_3]\cdot[Alkene]$ source term and $[SA]\cdot CS+\beta\cdot[SA]^2$ sink term do not show a strong correlation (Fig. 5 (d)). Most likely, this suggests that the source term cannot fully represent the actual SA source for heavy polluted conditions. For instance, there are likely additional sources of SA, such as direct emission from diesel vehicles, oil refineries, SA plants, and any other factories that use coal as heating or power supply (Srivastava et al., 2004;Arnold et al., 2006;Ahn et al., 2011;Roy et al., 2014;Sarnela et al., 2015;Godunov et al., 2017). Another possible cause for the correlation deviation under polluted condition, and to some extent also mildly polluted condition, is that the distribution of alkenes may not be constant for measurements classified into the same pollution level, that is, the $k_{app}$ is not constant. At very clean nights when alkenes sources are considered more local and stable, dramatic changes in alkene distribution are not expected. Moreover, our instrument is only capable of measuring a limited amount of alkene species and the fitted parameter $k_{app}$ might be overestimated.

It should be pointed out that we are not able to further deconvolute the contribution of OH radical and sCI based on the ambient observation. A well-tuned box model is a useful tool to resolve it and verify the role of the ozonolysis of alkenes on the nighttime SA formation. However, such a modeling work is not included in our study, as the lacking of a complete VOC datasets in our measurement and the largely uncertain yields of sCI from the ozonolysis of various alkenes have caused challenges in ensuring the precision of the box-model.

**4.4 Atmospheric Implication: Contribution of Nighttime Sulfuric Acid to Sub-3nm Particles**

We show that the ozonolysis of alkenes is the major source for the considerable amount of SA that exists at night, at least under unpolluted conditions. And it is found that increasing SA concentration coincided with increasing number concentration of sub-3nm particles (Fig. 6), suggesting that SA had a strong enhancement in the formation of newly formed particles, which is consistent with previous study (Cai et al., 2017). Different from SA, there was a negative correlation between the concentration of highly oxygenated organic molecules (HOMs) and $N_{Sub-3nm}$ for both nighttime and daytime (see Fig. S8 in the supplement), indicating that HOMs were not the main driver for the formation of sub-3nm particles. Then, these elevated SA concentration has the dominant contribution to the formation of sub-3nm particles in the nighttime of winter Beijing. This phenomenon is in contrast with some previous observations in forested areas where oxidation products of biogenic VOCs, especially monoterpene, were the main contributor to the formation of clusters (Eerdekens et al., 2009;Lehtipalo et al., 2011;Kammer et al., 2018;Rose et al., 2018). Both observation (Rose et al., 2018) and laboratory experiment (Lehtipalo et al., 2018) have shown that HOM dimers with extremely low volatility play a key role in the initial formation of clusters; however at our site in urban Beijing, high level of $NO_x$ inhibited the production of HOM dimers and most HOMs are monomers, which have minor importance to the formation of sub-3nm particles. Nighttime sub-3nm particles has also been observed at suburban site (Kecorius et al., 2015) or areas which are strongly influenced by coastal air masses (Yu et al., 2014;Salimi et al., 2017), but the underlying mechanism are still unclear.

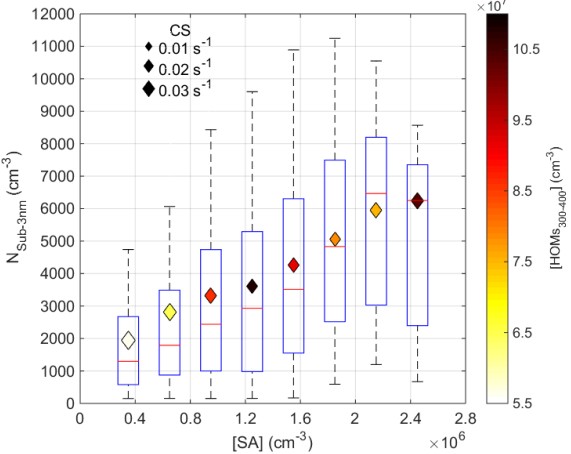

**Fig. 6** Correlation between $N_{Sub-3nm}$ and SA concentration during nighttime (20:00-04:00) from 18th January to 16th March 2019. The binned diamonds are colored by number concentration of HOMs (m/Q = 300 - 400 Th) and the size is related to CS. The red line is the median value, the bottom and the top blue lines are the 25 and 75 percentiles, and the whisker ranges cover the ± 2.7σ of those data in each bin.

**5. Conclusions**

Continuous SA measurement was conducted during the heating-supply period in urban Beijing. Frequent nighttime SA events were found and accounted for about 32 % of the total measurement nights. Most nighttime SA events were observed under unpolluted conditions and associated with a distinct drop of CS. We show that the SA source corresponding to the

product of $O_3$, alkenes and $SO_2$ concentrations correlates well with the SA sink for clean conditions, and to some extent also for mildly polluted conditions. Therefore, we suggest that nighttime SA formation under these conditions can be largely attributed to the ozonolysis of alkenes which leads to the production of OH radicals as well as sCI that are able to act as oxidants for $SO_2$. However, further deconvolution of the contribution of OH radicals, sCI and each possible alkene precursor was not possible within this study due to the inability to directly measure OH, sCI and the entire range of alkene precursors. It should also be pointed out that, under polluted conditions, there were very likely additional SA sources other than the ozonolysis of alkenes, such as direct emission from diesel vehicles, oil refineries and SA plants. Furthermore, we showed that these elevated SA had a dominant contribution to the formation of sub-3nm particles in the nighttime of winter Beijing.

## Acknowledgements

This project has received funding from the National Natural Science Foundation of China (41877306), the ERC advanced grant No. 742206, the European Union's Horizon 2020 research and innovation program under grant agreement No. 654109, the Academy of Finland Center of Excellence Project No. 27204. Kaspar R. Daellenbach acknowledges the support by the SNF mobility grant P2EZP2_181599.

**Data availability:** Data and codes are available upon asking the corresponding author (chao.yan@helsinki.fi)

**Author contribution:** YG, CY designed the study; YG, CY, CL, YZ, LD collected the data and materials; YG, CY, DS wrote the manuscript; all coauthors contributed to the scientific discussion and commented on the manuscript.

**Competing interests:** The authors declare no competing interests.

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
