# Peer review of "Formation of Nighttime Sulfuric Acid from the Ozonolysis of Alkenes in Beijing"

_Atmospheric Chemistry and Physics, 2019_

## Referee Comment (RC1) · Anonymous Referee #2 · 29 May 2020

This paper presents gas phase sulfuric acid measurements from Beijing during winter and summer with CIMS and show some nighttime formation of sulfuric acid, possibly from sCI formed from VOCs ozonolysis reactions. Sulfuric acid measurements are valuable –considering roles of sulfuric acid on new particle formation especially in urban environments. But data analysis is not effective – please see comments below. The other conclusion is nighttime sulfuric acid is responsible for sub-3 nm particles. How did the authors exclude HOMs from sub-3 nm particle formation?

Section 4.3. I have never seen a kinetic expression like this: $k_{app} \cdot [Alkene] \cdot [O_3] \cdot [SO_2]$. This does not make sense to me. Please show how $k_{app}$ is the same as $k_1 \cdot k_2 \cdot \varphi Âůf$? And what about $k_3$, $k_4$ and $k_5$ then? Does this mean nighttime sulfuric acid is not from OH + $SO_2$ reaction (OH from VOCs ozonolysis)? The authors assume

$\varphi$·f to be 0.05 – these are random numbers. The dimer formation rate betta is 3.46e-10 cmˆ3 sˆ-1 – isn't it too high? And what about the sink due to formation sulfuric-amines or sulfuric acid-ammonia clusters? And if this VOCs does not include monoterpenes and isoprene (emitted from biogenic emissions in summer and from volatile personal products year around), then how is the source-sink discussion really useful? I suggest the authors make steady state calculations of sCI and OH (or use a box model) to simulate nighttime sulfuric acid concentrations.

Please change Figure 1 in linear scale (Y axis) – as opposed to log scale, as a more common practice in the field (or adding inset in log scale). If I look at Figure s1 (move this to the main text or replace Figure 1 with this), it seems that nighttime sulfuric acid is quite minor. So I am not sure with the conclusion of nighttime sulfuric acid formation. Did authors measure only gas phase monomer sulfuric acid? Does this also include the sulfuric acid-amine cluster? Do they have measurements of dimer, trimer, tetramer of sulfuric acid, ammonia and amines during this period to show that sub-3 nm particles are really from sulfuric acid clusters? Please also include size distributions to show if there are new particle formation or not during the night.

Figure 5. Please also show HOMs in the same way as sulfuric acid, and include daytime data as well (vs. nighttime). Did sub-3 nm particles grow further? If they did not grow larger, then what are the possible explanations? Can you calculate J from PSM? What is p (power dependence) of J or sub-3 nm particles on nighttime sulfuric acid (vs. daytime)? Is p different during day and night?

Line 40: Needs refs., e.g., [Lee et al., 2019].

Line 43: Please include [Yu et al., 2012].

Line 51: please include [Erupe et al., 2010] and [Yu et al., 2013]

Line 54: needs refs. Is this statement true? I hardly see e7 cmˆ-3 level of sulfuric acid.

Line 60: please include [Yu et al., 2013].

Line 64: Change "frequent and noticeable" to "noticeable nighttime sulfuric acid some-times".

Line 105: This is a very long inlet. What is the residence time, and radius? Wall loss rate in the inlet?

Line 134: Calibration once a month? This is really infrequent! How frequently did they make background measurements?

Line 143/165: Regardless, SO2 seemed to me always at the ppb in average, so in high SO2 conditions year around.

Figure 2. Please show amines and ammonia.

Line 191-192: This does not make sense to me.

Figure s1: move this to the main text – including spring measurements.

Figure s3: the same as Figure 3?

Erupe, M. E., et al. (2010), Correlation of aerosol nucleation rate with sulfuric acid and ammonia in Kent Ohio: an atmospheric observation, J. Geophys. Res., 115, Doi:10.1029/2010JD013942, doi:10.1029/2010JD013942.

Lee, S.-H., H. Gordon, H. Yu, K. Lehtipalo, R. Haley, Y. Li, and R. Zhang (2019), New Particle Formation in the Atmosphere: From Molecular Clusters to Global Climate, Journal of Geophysical Research: Atmospheres, 124(13), 7098-7146, doi:10.1029/2018JD029356.

Yu, H., et al. (2013), Sub-3 nm particles observed at the coastal and continental sites in the United States, J. Geophys. Res., 119, Doi: 10.1029/2013JD020841, doi:10.1029/2013JD020841.

Yu, H., R. McGraw, and S. H. Lee (2012), Effects of amines on formation of sub-3 nm particles and their subsequent growth, Geophys. Res. Lett., 39, Doi:

10.1029/2011gl050099, doi:10.1029/2011gl050099.

---

## Referee Comment (RC2) · Anonymous Referee #1 · 23 Jun 2020

This paper shows that, within the suite of measurements they present, the nighttime formation of SA is consistent with a simplified chemistry driven by alkene, ozone, and SO2. Yet correlation does not constitute proof. Furthermore the procedures and details of the methodology (which may be correct) are either only sketched-out or are hard to follow (thus this reader did not have full confidence in the material.) Furthermore, the authors supply caveats (more than one time and even in the abstract!) that their analysis could be subject to revision/flawed. Providing a detailed, time-dependent simulation (even a box model) would bring their conclusion into the firmly believable realm. Below are some details and other points. Note the revisions are too strongly suggested as 'major': they are by no means damning and they should not be difficult to include or address. \

1) Somewhat careless with precision, quoting a four significant figure k_app from a slope that has at most two significant figures. A minor detail of course, but attention to detail should be demonstrated in all aspects. A welcome detail here would be to present the uncertainties in the values of the fitted slopes. \

2) Overall sink was equated to the alkene O3 SO2 source but steady-state assumption was not fully discussed (not even sure what time period the data is averaged over?). Furthermore SO3 to SA was not discussed. OH produced in alkene + O3 reactions was not included: why not? A simple box model could include these and others such as HO2 + NO for example. Then presenting box model simulations absent the alkene-ozone chemistry might really draw a distinct comparison. \

3) Is not sub-3 nm really sub-2.45 nm? \

4) The last figure purports to correlate sub-3 nm to measured SA and there is a linear relationship provided. Two problems: large error bars (what do they mean?) and there is a source of particles at zero SA (or zero alkene+ozone). The correlation may be due to the fact that the ordinate and abscissa are both dependent on the alkene-O3 chemistry yet sub-3 nm particles at 'zero' chemistry destroys the happiness of the association between ordinate and abscissa. Another issue is the lack of discussion regarding any proposed theoretical relationship between SA and number of particles. \

5) The outliers are numerous in many of the plots in Fig. 2. How were they decided upon? In this vein it is not clear what data was included for each of the points in Fig. 4 for example. All data between 10pm and 4am?

6) "SIZE = CS*xyz" was included in many of the plots but a reference size was not easy to find.

7) " calibration coefficient " has no meaning by itself. Needs some context (an equation) and perhaps some comparisons. It can be argued that this quantity should have units

of Hz Hz-1 attached to it also.

8) Why have PM2.5, visibility and CS all plotted in Fig. 2? Figure 2 would be cleaner if you pick one and plot the correlation between it and the others in the supplement....
* * *

---

## Author Comment (AC1) · 11 Aug 2020

**Response to Reviewers**
**Referee #1**

This paper shows that, within the suite of measurements they present, the nighttime formation of SA is consistent with a simplified chemistry driven by alkene, ozone, and $SO_2$. Yet correlation does not constitute proof. Furthermore, the procedures and details of the methodology (which may be correct) are either only sketched-out or are hard to follow (thus this reader did not have full confidence in the material.) Furthermore, the authors supply caveats (more than one time and even in the abstract!) that their analysis could be subject to revision/flawed. Providing a detailed, time-dependent simulation (even a box model) would bring their conclusion into the firmly believable realm. Below are some details and other points. Note the revisions are too strongly suggested as 'major': they are by no means damning and they should not be difficult to include or address.

We thank the reviewer for the constructive comments and suggestions. As suggested, we have added more details of the methodology to make our analysis easier to follow. We did not quite follow the comment on "the authors supply caveats (more than one time and even in the abstract!) that their analysis could be subject to revision/flawed", but we hope our point-to-point response to the comments as given below can address these concerns. The comments, our replies, and the corresponding changes in the manuscript and supplementary information are marked in black, blue, and green, respectively.

1) Somewhat careless with precision, quoting a four significant figure $k_{app}$ from a slope that has at most two significant figures. A minor detail of course, but attention to detail should be demonstrated in all aspects. A welcome detail here would be to present the uncertainties in the values of the fitted slopes.

Response: Thanks a lot for your suggestions and we have revised the $k_{app}$ values with two significant figures.
    The fitted value, 95% confidence bounds, uncertainty of $k_{app}$ and correlation coefficient $R^2$ for Clean-1 (Fig. 5(a)) and Clean-2 (Fig. 5(b)) condition are listed in Table S3. The uncertainties of $k_{app}$ are 20.2% and 11.3% for Clean-1 and Clean-2 condition respectively.
    For nighttime correlation between source term and sink term in Section 4.3 and Fig. 5, we mentioned in the manuscript that only under Clean-2 condition, there was a good correlation ($R^2$=0.97) for the mean values in all bins (Fig. 5 (b), Line 229-231, Page 9). For Clean-1 condition, there was only a subgroup of binned data with a source range from $1.0 \times 10^{33}$ to $4.5 \times 10^{33}$ (cm$^{-3}$)$^3$ that showed linear correlation (Line 223-224, Page 9). And for heavy polluted condition, no correlation was observed. (Line 253-254, Page 10).
    We have also added one sentence in the manuscript to refer to this table. "The fitted value, 95% confidence bounds, uncertainty of $k_{app}$ and correlation coefficient $R^2$ for Clean-1 and Clean-2 condition are listed in Table S3." (Line 233-234, Page 9).

**Table S3** Fitted value, 95% confidence bounds, uncertainty of $k_{app}$ and $R^2$ for Clean-1 and Clean-2 condition.

| Condition | $k_{app}$ (cm$^6$ s$^{-1}$) | 95% Confidence Bounds (cm$^6$ s$^{-1}$) | Uncertainty (%) | $R^2$ |
|---|---|---|---|---|
| Clean-1 | $2.7 \times 10^{-30}$ | ($2.1 \times 10^{-30}$ - $3.2 \times 10^{-30}$) | 20.2 | 0.97 |
| Clean-2 | $2.6 \times 10^{-30}$ | ($2.3 \times 10^{-30}$ - $2.9 \times 10^{-30}$) | 11.3 | 0.97 |

2) Overall sink was equated to the alkene $O_3$ $SO_2$ source but steady-state assumption was not fully discussed (not even sure what time period the data is averaged over?).

Response: The verification of the steady-state assumption is indeed necessary. We have added the following content in Section S2 to clarify why steady-state assumption can be assumed.
    The net concentration change of gaseous SA is determined by both the source and loss terms, as shown in the following equation:

$$\frac{d[SA]}{dt} = k_{app} \cdot [Alkene] \cdot [O_3] \cdot [SO_2] - [SA] \cdot CS - \beta \cdot [SA]^2$$

We can compare the magnitude of the net concentration change to the overall loss rate. During nighttime (20:00-04:00) from 18$^{th}$ January to 16$^{th}$ March 2019, the median net concentration change of SA is about 181.60 cm$^{-3}$s$^{-1}$ and the overall SA loss rate at the median SA concentration ($7.52 \times 10^5$ cm$^{-3}$) is $1.61 \times 10^4$ cm$^{-3}$s$^{-1}$. As the loss rate (and source rate) is much faster than the net concentration change, the pseudo-steady state (PSS) assumption is valid for SA. Besides, the resolution of the SA data is 5 minutes, and the concentration, net concentration change, loss rate and production rate of SA are listed in Table S4 below.

**Table S4** Concentration, net concentration change, loss rate and production rate of SA during nighttime (20:00-04:00) from 18$^{th}$ January to 16$^{th}$ March 2019. Std means standard deviation.

|  | [SA] (cm$^{-3}$) | d[SA]/dt (cm$^{-3}$s$^{-1}$) | $L_{SA}$ (cm$^{-3}$s$^{-1}$) | $P_{SA}$ (cm$^{-3}$s$^{-1}$) |
|---|---|---|---|---|
| Median | $7.52\times10^{5}$ | 181.60 | $1.61\times10^{4}$ | $1.60\times10^{4}$ |
| 25 percentile | $5.19\times10^{5}$ | 78.37 | $4.83\times10^{3}$ | $4.80\times10^{3}$ |
| 75 percentile | $1.05\times10^{6}$ | 371.14 | $3.27\times10^{4}$ | $3.26\times10^{4}$ |

To clarify this consideration, we added the above explanation in the revised supplement as Section S2 (Line 16-25, Page 1) and the following sentence in the revised manuscript:

"Under pseudo steady-state (PSS) assumption (see Section S2 for detailed disscussion about PSS assumption)" (Line 216-217, Page 8)

The time periods of the data in Fig. 5 and Fig. 6 are both from 18$^{th}$ January to 16$^{th}$ March 2019. Corresponding illustrations were added to the captions of Fig. 5 (Line 249, Page 10) and Fig. 6 (Line 285, Page 11).

3) Furthermore, SO$_3$ to SA was not discussed.

Response: Indeed, we did not discuss the conversion from SO$_3$ to SA, as the reaction is so fast that almost all SO$_3$ should be instantaneously converted to SA. To be more specific, the conversion of SO$_3$ to SA is based on the following reaction:

$$SO_3 + 2H_2O \xrightarrow{k} H_2SO_4 + H_2O$$

Then, the production rate of SA from SO$_3$ can be expressed as:

$$P_{[SA]} = k \cdot [SO_3] \cdot [H_2O]^2$$

where k=$3.9\times10^{-41}$ exp(6830.6/T) cm$^{-6}$ s$^{-1}$ ((Jayne et al., 1997)). During the nights of the measurement period, the median concentration of H$_2$O was $4.97\times10^{16}$ cm$^3$, and the median temperature was 276.6 K, then the lifetime of SO$_3$ can be estimated as:

$$\tau_{SO_3} = \frac{1}{k \cdot [H_2O]^2} = \frac{1}{4.88 \times 10^3 \ s^{-1}} = 2.05 \times 10^{-4} \ s$$

As the lifetime of SO$_3$ is so short under typical atmospheric conditions, the oxidation of SO$_2$ is the rate-limiting step in the formation of SA.

To clarify this consideration, we added the above illustration in Section S1 (Line 2-14, Page 1) and the following explanation in the revised manuscript:

"The lifetime of SO$_3$ due to its fast reaction with H$_2$O to form SA is approximately $2\times10^{-4}$ s (see detailed calculation in supplement Section S1), which indicates that this reaction is so fast that almost all SO$_3$ will be instantaneously converted to SA. In this case, the oxidation of SO$_2$ is the rate-limiting step in the formation of SA." (Line 84-86, Page 3)

4) OH produced in alkene + O$_3$ reactions was not included: why not?

Response: This is a misunderstanding. We didn't mean to exclude the non-photochemical ·OH oxidation pathway. Actually, we do not attempt to separate the contributions of sCI and OH radical in this study, but instead, use the k$_{app}$ as an empirical parameter to account for the overall effect of sCI and OH oxidation pathways.

We realized that the comparison between k$_{app}$ and theoretical sCI oxidation rate is confusing that caused the misunderstanding. Besides, such comparison also suffers from large uncertainties. Therefore, we decide to remove such discussion in the revised manuscript so that the main message can be clearer. However, we need to point out that, this change in the manuscript will not change our conclusion that the ozonolysis of alkenes is responsible for the oxidation reactions (both sCI and non-photochemical OH radical) that drive the nighttime SA formation.

To clarify this consideration, the following illustration are added to the revised manuscript:

"The ozonolysis of alkenes under dark conditions is capable of generating sCI as well as OH radical, both of which are able to oxidize SO$_2$ to form gaseous SA. However, the yields of both sCI and recycled OH radical remain largely unquantified. Therefore, we do not attempt to distinguish the contribution of sCI and OH radical on SA formation in this study, but rather treat them as a "bulk oxidant" and use an empirical parameter k$_{app}$ to account for both oxidation pathways." (Line 205-209, Page 8)

And the following discussion are deleted from the revised manuscript:

"In order to have a general understanding of the apparent rate constant of sCI-SO$_2$ reaction obtained from our measurement, we can roughly get an upper limit value by considering all nighttime SA is produced from the sCI mechanism. From the above discussion, the slope k$_{app}$ can be expressed as k$_1$·k$_2$·φ·f, where f is the fraction of sCI which undergo the reaction with SO$_2$. It should be pointed out that k$_2$ is also an apparent rate constant which results from the combination of different measurement efficiency of alkenes (including undetected ones), different yields of sCI, and different rate constants of sCI reacting with SO$_2$. The fitted k$_{app}$ is $2.618\times10^{-30}$ cm$^6$ s$^{-2}$. If considering k$_1$ to be $1.0\times10^{-17}$ cm$^3$s$^{-1}$ (an intermediate value in the range of previous

studies, which has been explained in Section 2), then $k_2 \cdot \varphi \cdot f = 2.618 \times 10^{-13}$ cm$^3$s$^{-1}$. As the real atmospheric chemical composition is far more complex than experimental ones, the value of $\varphi \cdot f$ should be smaller. Thus, if further considering $\varphi \cdot f$ to be 0.05, then the rate constant $k_2$ in this real atmospheric condition is approximate $5.236 \times 10^{-12}$ cm$^3$s$^{-1}$, which is in the same order of magnitude as measured values in experiments (see Section 2)." (between Line 234 and Line 235, Page 9)

5) A simple box model could include these and others such as HO$_2$ + NO for example. Then presenting box model simulations absent the alkene-ozone chemistry might really draw a distinct comparison.

Response: Performing a well-tuned box model is indeed a useful way of verifying our findings. However, after thinking it through, we found that many species and parameters needed for the box model remain unquantified or largely uncertain. For example, the concentration of OH radical in the nighttime is one key parameter in such a box model, but lacking of a complete VOC measurement by GCMS will greatly limit the precision of the box model. And recent study also showed that current MCM method significantly under-predicts the concentrations of OH, HO$_2$ and RO$_2$ radicals. (Slater et al., 2020) Besides, the yields of sCI and OH radical as well as the rate constants concerning R1 to R5 of alkenes are largely scattered (Line 90-94 Page 3). Therefore, we prefer not to include the box model simulation in this manuscript, but clearly state the need for a box model based on further lab studies to fully to verify our results.
    To clarify this consideration, the following discussion has been added to the revised manuscript:
    "It should be pointed out that we are not able to further deconvolute the contribution of OH radical and sCI based on the ambient observation. A well-tuned box model is a useful tool to resolve it and verify the role of the ozonolysis of alkenes on the nighttime SA formation. However, such a modeling work is not included in our study, as the lacking of a complete VOC datasets in our measurement and the largely uncertain yields of sCI from the ozonolysis of various alkenes have caused challenges in ensuring the precision of the box-model." (Line 263-267, Page 10)

6) Is not sub-3 nm really sub-2.45 nm?

Response: Such a particle size range is determined by the instrument (PSM). According to the calibration of the instrument, the size bins of PSM are 1.3-1.44 nm, 1.44-1.5 nm, 1.5-1.61 nm, 1.61-1.81 nm and 1.81-2.45 nm, and the number concentration of sub-3nm particles is the sum of all those five size bins. Uncertainties in response of PSM to particles of different chemical composition (Kangasluoma et al., 2014) can shift the overall size-range where the PSM is sensitive, which is the reason why the total concentration measured between the lowest and highest size in PSM is generally referred as sub-3 nm particle concentration (see e.g. (Kontkanen et al., 2017)).
    To clarify this consideration, the following sentence has been added to the revised manuscript:
    "(…) was measured with a Particle Sizer Magnifier (PSM) (Vanhanen et al., 2011), and the integrated number concentration of particles from PSM is referred as N$_{Sub-3nm}$ in the following" (Line 139-140, Page 5)

7) The last figure purports to correlate sub-3 nm to measured SA and there is a linear relationship provided. Two problems: large error bars (what do they mean?) and there is a source of particles at zero SA (or zero alkene+ozone). The correlation may be due to the fact that the ordinate and abscissa are both dependent on the alkene-O$_3$ chemistry yet sub-3 nm particles at 'zero' chemistry destroys the happiness of the association between ordinate and abscissa. Another issue is the lack of discussion regarding any proposed theoretical relationship between SA and number of particles.

Response: The large error bars indicate the scattering of those data points in each bin. For better illustration, we revised this plot with boxplots in SA bins (see below), which provide direct information on the data distribution. In the boxplot, the red line is the median value, the bottom and top blue lines are the 25 and 75 percentiles, and the whisker ranges cover the $\pm$ 2.7$\sigma$ of those data in each bin. In addition, the mean values are added as diamonds.

[Figure]

**Original Fig. 5** Nighttime correlation between $N_{Sub-3nm}$ and SA concentration during nighttime (20:00-04:00) from 18th January to 16th March 2019. The data points are colored by number concentration of HOM (m/Q = 300 - 400 Th) and the size is related to CS. Note that the data points are based on the binned data instead of the original one.

[Figure]

**Updated Fig. 5** Correlation between $N_{Sub-3nm}$ and SA during nighttime (20:00-04:00) from 18th January to 16th March 2019. The binned diamonds are colored by number concentration of HOMs (m/Q = 300 - 400 Th) and the size is related to CS. The red line is the median value, the bottom and the top blue lines are the 25 and 75 percentiles, and the whisker ranges cover the ± 2.7σ of those data in each bin.

To clarify this consideration, we replaced the original Fig. 5 with the updated one in the manuscript. Please also note that Fig. 5 becomes Fig. 6 in the revised manuscript.

The particles at zero SA were most likely from other sources than SA. As we mentioned, our measurement site is close to two main urban traffic trunk roads, and these particles might come from directly emission of vehicles (Arnold et al., 2006;Barrios et al., 2012). In a recent study, it is shown that PSM is very sensitive to traffic-emitted sub-3nm particles (Ronkko et al., 2017).

Regarding the theoretical relationship between number concentration of sub-3nm particles and SA concentration, it is usually done by depicting the particle nucleation rate (J, e.g., $J_{1.5}$, $J_{1.7}$) as a function of SA concentration, and comparing it with other chamber or ambient studies with known mechanisms. Calculation from number concentration (N) to J involves many corrections including the correction of particle growth out of the size range. However, in these nighttime SA events, the determination of particle growth rate is challenging as the "banana shape" is not clear (please also see the reply to comment #2) of the other reviewer). This is the main reason why we used the number concentration, as a more objective term, instead of the calculated particle nucleation rate in this study.

However, to address the reviewer's concern, we estimated J by ignoring the growth rate correction term in the calculation. Fig. R1 shows the correlation between $J_{1.5}$/$J_{1.7}$ and SA for Beijing measurement, Shanghai measurement (Almeida et al., 2013) and CLOUD experiments (Almeida et al., 2013;Kirkby et al., 2011). As shown in Fig. R1, data points in the nighttime and the

daytime are roughly falling on the same line, which also agree well with the data measured in Shanghai CLOUD chamber SA-DMA-$H_2O$ experiment. The similar J − SA relationship between the nighttime and the daytime suggests a similar nucleation mechanism as SA-base clustering. However, as the calculation of J has the aforementioned uncertainty, we prefer not to include Fig. R1 in the manuscript.

[Figure]

**Fig. R1** Comparison of Beijing ambient, Shanghai ambient and CLOUD experimental cluster formation rates against SA concentration. Green, light blue and grey dots denote CLOUD $J_{1.7}$ data for SA-$H_2O$, SA-$NH_3$- $H_2O$ and SA-DMA- $H_2O$ nucleation respectively (Almeida et al., 2013;Kirkby et al., 2011). Magenta diamonds represent Shanghai NPF $J_{1.7}$ data (Yao et al., 2018). Red and blue diamonds are Beijing $J_{1.5}$ data for NPF day (10:00-14:00) and Clean-2 night (20:00-04:00), respectively.

Furthermore, there was a negative correlation between $N_{Sub-3nm}$ and highly oxygenated organic molecules (HOMs) (see Fig. S8 (a) below), which indicates that HOMs was not the main driver for the formation of nighttime sub-3nm particles. And to clarify this consideration, we added Fig. S8 (a) in the revised supplement.

[Figure]

**Fig. S8** (a) Correlation between $N_{Sub-3nm}$ and [HOMs] during nighttime (20:00-04:00) from 18th January to 16th March 2019. The grey dots are original data points. The diamonds are binned data colored by number concentration of SA and the size is proportional to CS. The blue lines are standard deviation of data points in each bin.

8) The outliers are numerous in many of the plots in Fig. 2. How were they decided upon? In this vein it is not clear what data was included for each of the points in Fig. 4 for example. All data between 10 pm and 4 am?

Response: Please note that now Fig. 2 is Fig. 3, and Fig. 4 is Fig. 5.

The outliers in Fig. 3 are the ones out of the ± 2.7σ range of all selected data. If the data is normally distributed, this ± 2.7σ range will cover 0.7 - 99.3 percentiles of the data. Corresponding illustrations have been added to the captions of Fig. 3 (Line 186, Page 7) in the revised manuscript.

Data points in Fig. 5 are the ones during nighttime (20:00-04:00) from 18[th] January to 16[th] March 2019. Corresponding illustrations have been added to the captions of Fig. 5 (Line 249, Page 10). Fig. 5 (a), (b), (c) and (d) are for Clean-1, Clean-2, mildly polluted and heavy polluted conditions respectively. The definition of Clean-1, Clean-2, mildly polluted and heavy polluted conditions have been illustrated in the manuscript (Line 199-201, Page8 and Line 226-227, Page 9). For better understanding of Fig. 5, the legends in four subplots have been changed from 'Vis ≥ 12 km', 'Vis ≥ 16 km, $[O_3] ≥ 2×10^{11}$ cm$^{-3}$', 'Vis: 4-12 km' and 'Vis ≤ 4 km' to 'Clean-1', 'Clean-2', 'Mildly polluted' and 'Heavy polluted' accordingly.

9) "SIZE = CS*xyz" was included in many of the plots but a reference size was not easy to find.

Response: Thanks a lot for your suggestions. We've added three CS references points (CS=0.01 s$^{-1}$, 0.02 s$^{-1}$ and 0.03 s$^{-1}$) in all relevant plots.

10) " calibration coefficient " has no meaning by itself. Needs some context (an equation) and perhaps some comparisons. It can be argued that this quantity should have units of Hz Hz-1 attached to it also.

Response: Thanks for your suggestion. We have now added the following equation to clarify how SA concentration is calculated.

The quantification of sulfuric acid is derived from the ratio of bisulfate ions relative to primary ions as follows:

$$[H_2SO_4] = \frac{HSO_4^- + H_2SO_4NO_3^-}{NO_3^- + HNO_3NO_3^- + (HNO_3)_2NO_3^-} × C$$

The calibration factor, C, is determined from direct calibration where gaseous sulfuric acid of known amounts is produced and injected into the instrument. A more detailed information about the calibration is discussed by Kürten et al. 2012 (Kurten et al., 2012) . The units of bisulfate and primary ions are both counting rates in ions·s$^{-1}$ and cancel each other, and therefore, the unit of C is the same as that of sulfuric acid concentration in cm$^{-3}$.

To clarify this consideration, we added the following illustration in the revised manuscript:

"The quantification of sulfuric acid is derived from the ratio of bisulfate ions (with counting rates unit in ions·s$^{-1}$) relative to primary ions as follows:

$$[H_2SO_4] = \frac{HSO_4^- + H_2SO_4NO_3^-}{NO_3^- + HNO_3NO_3^- + (HNO_3)_2NO_3^-} × C$$

The calibration factor, C, is determined from direct calibration by injecting gaseous sulfuric acid of known amounts into the instrument (Kurten et al., 2012)." (Line 114-118, Page 4)

11) Why have PM2.5, visibility and CS all plotted in Fig. 2? Figure 2 would be cleaner if you pick one and plot the correlation between it and the others in the supplement....

Response: Thanks for your suggestion and please note that Fig. 2 now is Fig. 3. The correlation among these three parameters have been shown in Fig. S3.

From the perspective of cleanliness, picking one parameter among PM$_{2.5}$, visibility and CS, and moving the correlation figure to the supplement is reasonable. PM$_{2.5}$ is the most commonly used parameter to describe pollution level, visibility is used to distinguish pollution level in Section 4.3, and nighttime SA events were highly associated with CS level (Table S2). As all these parameters are useful for the later discussion, we are prone to keep them in the current form.

---

## Author Comment (AC2) · 11 Aug 2020

**Referee #2**

This paper presents gas phase sulfuric acid measurements from Beijing during winter and summer with CIMS and show some nighttime formation of sulfuric acid, possibly from sCI formed from VOCs ozonolysis reactions. Sulfuric acid measurements are valuable – considering roles of sulfuric acid on new particle formation especially in urban environments. But data analysis is not effective – please see comments below. The other conclusion is nighttime sulfuric acid is responsible for sub-3 nm particles. How did the authors exclude HOMs from sub-3 nm particle formation?

We thank the reviewer for the constructive comments and suggestions and we have carefully revised our manuscript and supplement accordingly. The point-to-point response to the comments is given below. The comments, our replies, and the corresponding changes in the manuscript and supplementary information are in black, blue, and green, respectively. In this study, HOMs are not likely the determining species for the formation of sub-3nm particles and detailed discussions are shown in the response to question '3)'.

1) Section 4.3. I have never seen a kinetic expression like this:  $k_{app} \cdot [Alkene] \cdot [O_3] \cdot [SO_2]$ . This does not make sense to me. Please show how  $k_{app}$  is the same as  $k_1 \cdot k_2 \cdot \varphi \cdot f$ ? And what about  $k_3$ ,  $k_4$  and  $k_5$  then? Does this mean nighttime sulfuric acid is not from OH + SO2 reaction (OH from VOCs ozonolysis)? The authors assume are really from sulfuric acid clusters?

Response: Both sCI and non-photochemical  $\cdot$ OH are capable of oxidizing SO2 to form SA at night. Actually, the contribution of sCI and OH radical cannot be distinguished in our study, and  $k_{app}$  expressed by  $k_1 \cdot k_2 \cdot \varphi \cdot f$  in the original manuscript was used to roughly get an upper limit of the sCI oxidation pathway by assuming the contribution of OH radical is negligible. But we found this part of discussion caused much confusion and decided to remove it from the revised manuscript.

The  $k_{app} \cdot [Alkene] \cdot [O_3] \cdot [SO_2]$  is an informal expression of describing the formation rate of SA, where  $k_{app}$  is an overall empirical parameter that takes into account the OH radical and sCI oxidation pathways resulted from the ozonolysis of alkenes. We use this empirical parameter because the detailed chemical formation pathways and corresponding parameters from alkenes, O3 and SO2 to SA are still not fully quantified yet.

To clarify this consideration, the following illustration are added to the revised manuscript:

"The ozonolysis of alkenes under dark conditions is capable of generating sCI as well as OH radical, both of which are able to oxidize SO2 to form gaseous SA. However, the yields of both sCI and recycled OH radical remain largely unquantified. Therefore, we do not attempt to distinguish the contribution of sCI and OH radical on SA formation in this study, but rather treat them as a "bulk oxidant" and use an empirical parameter  $k_{app}$  to account for both oxidation pathways." (Line 205-209, Page 8)

And the following discussion are deleted from the revised manuscript:

"In order to have a general understanding of the apparent rate constant of sCI-SO2 reaction obtained from our measurement, we can roughly get an upper limit value by considering all nighttime SA is produced from the sCI mechanism. From the above discussion, the slope  $k_{app}$  can be expressed as  $k_1 \cdot k_2 \cdot \varphi \cdot f$ , where f is the fraction of sCI which undergo the reaction with SO2. It should be pointed out that  $k_2$  is also an apparent rate constant which results from the combination of different measurement efficiency of alkenes (including undetected ones), different yields of sCI, and different rate constants of sCI reacting with SO2. The fitted  $k_{app}$  is  $2.618 \times 10^{-30}$  cm6 s-2. If considering  $k_1$  to be  $1.0 \times 10^{-17}$  cm3s-1 (an intermediate value in the range of previous studies, which has been explained in Section 2), then  $k_2 \cdot \varphi \cdot f = 2.618 \times 10^{-13}$  cm3s-1. As the real atmospheric chemical composition is far more complex than experimental ones, the value of  $\varphi \cdot f$  should be smaller. Thus, if further considering  $\varphi \cdot f$  to be 0.05, then the rate constant  $k_2$  in this real atmospheric condition is approximate  $5.236 \times 10^{-12}$  cm3s-1, which is in the same order of magnitude as measured values in experiments (see Section 2)." (between Line 234 and Line 235, Page 9)

The reviewer also asked that "The authors assume are really from sulfuric acid clusters?". Does the reviewer mean to ask that "Is the nighttime sub-3nm particles are formed from SA clusters?". And our answer is yes. As shown in Fig. 6 (please note that original Fig. 5 now is Fig. 6), there was a positive linear correlation between Nsub-3nm and SA, suggesting that SA was the main driving species for sub-3nm particles formation. Detailed discussion on the formation mechanism of sub-3nm particles is shown in response to comment '3)' and '4)'.

2) Please also include size distributions to show if there are new particle formation or not during the night.

Response: As particles measured by PSM only cover the size range of 1.3 nm to 2.45 nm, which is not able to show whether there is new particle formation or not, we plotted the size distribution of negative ions measured by neutral cluster and air ion spectrometer (NAIS) (Fig. R2). In general, among 57 nights from 18th January to 15th March 2019, there were 9 nights with elevated sub-3nm ions (marked with black rectangles in Fig, R2), but further growth of sub-3nm clusters was not observed.

---

## Author Response (AR2)

**Response to Reviewers**

We thank the reviewer for the constructive comments and suggestions. As suggested, we have added some details to make our analysis easier to follow and we hope our point-to-point response to the comments given below can address these concerns. The comments, our replies, and the corresponding changes in the manuscript and supplementary information are marked in black, blue, and green, respectively.

1) It looks to me that most of revisions were not made on the text or SI, though the authors tried to include additional figures to address comments in their response. I would like these figures to be included in revised manuscript.

Response: The previous changes made in the manuscript and supplement are marked in orange. Please note that in the following replies, the acronym SA stands for sulfuric acid, which has also been defined in the manuscript.

As suggested, Fig. 6 (b) and the corresponding illustration have been additionally added in the revised manuscript:

"Fig. 6 (b) shows that the particle nucleation rates (J) in the nighttime and the daytime roughly fall on the same line. Especially, the daytime values agree well with those from previous observations in Shanghai and CLOUD chamber SA-DMA (dimethylamine)-$H_2O$ experiment. The similar J – SA relationship between the nighttime and the daytime suggests a similar nucleation mechanism as SA-base clustering. Lei Yao et al. proposed that in urban megacities, high concentration of DMA together with SA were enough to explain the particle growth to ~3 nanometers (Yao et al., 2018). Therefore, the formation of nighttime as well as daytime sub-3nm particles at our site was more likely a result of SA-DMA-$H_2O$ nucleation." (Line 284-289, Page 11)

[Figure]

**Fig. 6 (b)** Comparison of Beijing ambient, Shanghai ambient and CLOUD experimental cluster formation rates against SA concentration. Green, light blue and grey dots denote CLOUD $J_{1.7}$ data for SA-$H_2O$, SA-$NH_3$-$H_2O$ and SA-DMA-$H_2O$ nucleation respectively (Almeida et al., 2013;Kirkby et al., 2011). Magenta diamonds represent Shanghai NPF $J_{1.7}$ data (Yao et al., 2018). Red and blue diamonds are Beijing $J_{1.5}$ data for NPF day (10:00-14:00) and Clean-2 night (20:00-04:00), respectively.

Besides, Fig. S9 and Table S5 have also been added to the revised supplement with the following explanation being added in the revised manuscript:

"Besides, João Almeida et al. suggested that when SA concentration doesn't exceed $3.0 \times 10^7$ cm$^{-3}$, the level of amines above 5 ppt are sufficient to reach the rate limit for amine ternary nucleation (Almeida et al., 2013). Although the median nighttime concentration of C2 amines (very likely DMA) was around 2.4 ppt (Fig. S9 and Table S5), there were also other base species (e.g., C3-amines, NH$_3$) co-existing. Altogether, they are sufficient to stabilize SA clusters. (Line 289-293, Page 11)

Please also note that Fig. 6 now is Fig. 6 (a).

[Figure]

**Fig. S9** Median diurnal variation of NH$_3$, C2 amines and C3 amines from 10$^{th}$ December, 2018 to 6$^{th}$ January, 2019.

**Table S5** Median concentrations of NH$_3$, C2 amines and C3 amines from 10$^{th}$ December 2018 to 6$^{th}$ January 2019.

| Species | Unit | Night (20:00-04:00) | Polluted Night (Vis < 12 km) | Clean-2 Night (Vis ≥ 16km, [O$_3$] ≥ 2×10$^{11}$ cm$^{-3}$) |
|---|---|---|---|---|
| NH$_3$ | Mixing ratio in ppb | 2.8 | 3.3 | 1.9 |
| | Concentration in cm$^{-3}$ | $7.6 \times 10^{10}$ | $8.9 \times 10^{10}$ | $5.1 \times 10^{10}$ |
| C2 Amines | Mixing ratio in ppt | 2.4 | 2.6 | 1.6 |
| | Concentration in cm$^{-3}$ | $6.3 \times 10^7$ | $7.0 \times 10^7$ | $4.4 \times 10^7$ |
| C3 Amines | Mixing ratio in ppt | 1.2 | 1.4 | 0.74 |
| | Concentration in cm$^{-3}$ | $3.1 \times 10^7$ | $3.9 \times 10^7$ | $2.0 \times 10^7$ |

2) Also, in a new ESTLett paper that came out recently (https://pubs-acs-org.elib.uah.edu/doi/10.1021/acs.estlett.0c00615), the authors showed SO3 formation (hence, sulfuric acid and even sub-3 nm particles) during the night due to heterogeneous reactions of SO2 on soot particles. Then the steady state of sulfuric acid must include this production route as well.

Response: Thanks a lot for your suggestion.

The ambient SO$_3$ was indeed observed. We first look into the correction between the measured concentrations of SO$_3$ and SA. As shown in Fig. S10 (a), under Clean-2 condition when the oxidation from alkenes ozonolysis pathway dominates the formation of SA, there is almost no apparent correlation between SA and SO$_3$. With the increase of pollution level, the positive correlation shows up. Under polluted condition, high aerosol surface could favor the heterogeneous production of SO$_3$, which might, at least to some extent, explain the extra SA sources. For this reason, we mentioned in the manuscript that there could be additional sources of SA under polluted condition.

It should be pointed out that, by assuming the pseudo-steady state, we calculated the concentration of SA based on the measured $SO_3$ data:

$$k_6[SO_3][H_2O]^2 = [SA]CS + \beta[SA]^2$$

where $k_6 = 3.9 \times 10^{-41} \exp(6830.6/T)$ cm$^{-6}$ s$^{-1}$ (Jayne et al., 1997). The calculated SA concentration was over 10 times higher than the measured one. As mentioned in the ESTLett paper, "$SO_3$ concentration could be overestimated because the influence of ambient ions (i.e., $SO_3 \cdot NO_3^-$) was not excluded and some of the hydrate complex intermediate ($SO_3 \cdot H_2O$) also could be detected as $SO_3 \cdot NO_3^-$ ". This means that not all measured $SO_3$ molecules/clusters were the ones producing SA. Due to the remaining uncertainties in the measured $SO_3$ concentration and the unclear mechanism of the $SO_2$ heterogeneous pathway, we are not able to quantify the formation of $SO_3$ and SA from the heterogeneous reactions of $SO_2$ on soot particles. Therefore, we can't include this $SO_2$ heterogeneous term in the production of SA.

[Figure]

**Fig. S10** Correlation between sulfuric acid monomer and $SO_3$ measured by CI-APi-TOF during nighttime (20:00 – 04:00 next day) from 18th January to 15th March 2019 for (a) Clean-2 condition with visibility larger than 16.0 km and $O_3$ concentration higher than $2.0 \times 10^{11}$ cm$^{-3}$ (~ 7 ppb), (b) Clean-1 condition with visibility larger than 12.0 km, (c) mildly polluted condition with visibility in the range of 4.0 - 12.0 km, and (d) heavy polluted condition with visibility smaller than 4.0 km.

To clarify this consideration, we added Fig. S10 in the revised supplement and the following interpretation in the revised manuscript:

"Besides, a recent study shows that $SO_3$ generated from the reaction of $SO_2$ on the surface of soot particles potentially leads to the formation of SA during nighttime and early morning (Yao et al., 2020). However, the correlation between the nighttime SA and $SO_3$ was only found during polluted periods (Fig. S10), suggesting that it might have an important contribution during polluted cases. This may, at least to some extent, explain the extra SA sources under polluted condition. (Line 256-260, Page 10)

3) Nevertheless, a simple 0-D box model simulations will be most useful.

Response: Thanks for your suggestion.

As suggested, the MCMv3.3.1 was used to simulate the production rate of SA ($P_{SA}$) from the alkenes ozonolysis route. In total, 26 VOCs, including 6 alkenes, and 5 trace gases were added as input parameters (Table R1). It should be noted that the isomers of VOCs molecules cannot be distinguished due to the lack of structure information, and fractions of different isomers

were roughly set to be equal. Both the sCI and OH pathways were simulated and their production reactions are listed in Table R2 and Table R3.

Fig. R2 shows that the median values of $P_{SA}$ for all nights (20:00-04:00) and Clean-2 nights during the measurement period (from 18$^{th}$ January to 16$^{th}$ March 2019) are $1.61\times10^4$ cm$^{-3}$ s$^{-1}$ and $6.07\times10^3$ cm$^{-3}$ s$^{-1}$ respectively. As mentioned in the manuscript, the rate constant between stabilized Criegee intermediate (sCI) and SO$_2$ ($k_{sCI\text{-}SO_2}$) spans over several orders of magnitude, from $7.0\times10^{-14}$ cm$^3$ s$^{-1}$ (MCMv3.3.1) to $4.0\times10^{-11}$ cm$^3$ s$^{-1}$ (Ahrens et al., 2014). As expected, with the increase of $k_{sCI\text{-}SO_2}$, the modelled $P_{SA}$ also increases dramatically, varying from $2.73\times10^3$ cm$^{-3}$ s$^{-1}$ to $5.61\times10^4$ cm$^{-3}$ s$^{-1}$. Although a uniform $k_{sCI\text{-}SO_2}$ can be adjusted to match the measured SA values, but it also reflects a large uncertainty of the model. Besides, there exist large deviations between the modelled and measured concentrations of SA, with daytime SA exceedingly higher than the measured one no matter what $k_{sCI\text{-}SO_2}$ value is chosen (Fig. R3), which further reveals the uncertainty in modeling $P_{SA}$ and SA concentration. Therefore, we still feel it is unnecessary to add the box model results in this study. However, we did mention in the manuscript that "A well-tuned box model is a useful tool to resolve it and verify the role of the ozonolysis of alkenes on the nighttime SA formation. However, such a modeling work is not included in our study, as the lacking of a complete VOC datasets and the large uncertainties in yields of sCI and oxidation rate constants of SO$_2$ by sCI have posed challenges in ensuring the precision of the box-model.". (Line 266-269, Page 10)

[Figure]

**Fig. R2** Production rate of SA ($P_{SA}$) from measurement and box model simulation. The 'All' and 'Clean2' represent dataset belonging to all night and clean-2 night. The 'k=7e-14', 'k=8e-13', 'k=4e-12', 'k=8e-12', and 'k=4e-11' are the reaction rate constants between sCI and SO$_2$ with unit of cm$^3$ s$^{-1}$. The rate constant of $7.0\times10^{-14}$ cm$^3$ s$^{-1}$ is chosen from MCMv3.3.1 as a lower limit. The rate constants of $8.0\times10^{-13}$ cm$^3$ s$^{-1}$ and $4.0\times10^{-11}$ cm$^3$ s$^{-1}$ are taken from literature (Mauldin et al., 2012;Ahrens et al., 2014) and latter one is regarded as the upper limit. The middle two $4.0\times10^{-12}$ cm$^3$ s$^{-1}$ and $8.0\times10^{-12}$ cm$^3$ s$^{-1}$ are used for sensitivity analysis for the box model. The black and colored values are the corresponding median values of $P_{SA}$.

[Figure]

**Fig. R3** Diurnal variations of measured and modelled SA concentration. The grey dashed line, red dashed line and grey dot-dashed line represent the 25 %, median and 75 % values of measured SA concentrations. The blue, green, orange, magenta and red solid lines are simulated SA concentrations from box model with different values of $k_{sCI-SO_2}$ with unit of $cm^3 \, s^{-1}$.

**Table R1** Input VOCs and trace gases for box model simulation

| Group | Name of Species | Formula |
|---|---|---|
| Trace Gases | Sulfur Dioxide | $SO_2$ |
| | Ozone | $O_3$ |
| | Carbon Monoxide | $CO$ |
| | Nitrogen Dioxide | $NO_2$ |
| | Nitric Oxide | $NO$ |
| Alkenes | Propylene | $C_3H_6$ |
| | Butadiene | $C_4H_6$ |
| | Butene | $C_4H_8$ |
| | Isoprene | $C_5H_8$ |
| | Pentene | $C_5H_{10}$ |
| | Hexene | $C_6H_{12}$ |
| Other VOCs | Butane | $C_4H_{10}$ |
| | Acetone | $C_3H_6O$ |
| | Pentane | $C_5H_{12}$ |
| | Methyl Ethyl Ketone | $C_4H_8O$ |
| | Hexane | $C_6H_{14}$ |
| | Methylcyclohexane | $C_7H_{14}$ |
| | Heptane | $C_7H_{16}$ |
| | Octane | $C_8H_{18}$ |
| | Nonane | $C_9H_{20}$ |
| | Decane | $C_{10}H_{22}$ |
| | Undecane | $C_{11}H_{24}$ |
| | Dodecane | $C_{12}H_{26}$ |

| | |
|---|---|
| Benzene | $C_6H_6$ |
| Toluene | $C_7H_8$ |
| Styrene | $C_8H_8$ |
| Xylene | $C_8H_{10}$ |
| Ethylbenzene | $C_8H_{10}$ |
| Trimethylbenzene | $C_9H_{12}$ |
| Ethyltoluene | $C_9H_{12}$ |
| Propylbenzene | $C_9H_{12}$ |

**Table R2** Production reactions of Criegee intermediate

| Species | Formula | Reactions |
|---|---|---|
| Propylene | $C_3H_6$ | $O_3 + C_3H_6 = CH_2OOB + CH_3CHO$ |
| | | $O_3 + C_3H_6 = CH_3CHOOA + HCHO$ |
| Butadiene | $C_4H_6$ | $O_3 + C_4H_6 = ACR + CH_2OOD$ |
| | | $O_3 + C_4H_6 = HCHO + ACROOA$ |
| 1-Butene | $C_4H_8$ | $BUT1ENE + O_3 = CH_2OOB + C_2H_5CHO$ |
| | | $BUT1ENE + O_3 = C_2H_5CHOOA + HCHO$ |
| Trans-2-Butene | | $TBUT2ENE + O_3 = CH_3CHO + CH_3CHOOB$ |
| Cis-2-Butene | | $CBUT2ENE + O_3 = CH_3CHO + CH_3CHOOB$ |
| Isoprene | $C_5H_8$ | $O_3 + C_5H_8 = CH_2OOE + MVK$ |
| | | $O_3 + C_5H_8 = CH_2OOE + MACR$ |
| | | $O_3 + C_5H_8 = HCHO + MVKOOA$ |
| | | $O_3 + C_5H_8 = HCHO + MACROOA$ |
| 1-Pentene | $C_5H_{10}$ | $PENT1ENE + O_3 = CH_2OOB + C_3H_7CHO$ |
| | | $PENT1ENE + O_3 = C_3H_7CHOOA + HCHO$ |
| Trans-2-Pentene | | $TPENT2ENE + O_3 = C_2H_5CHOOB + CH_3CHO$ |
| | | $TPENT2ENE + O_3 = CH_3CHOOB + C_2H_5CHO$ |
| Cis-2-Pentene | | $CPENT2ENE + O_3 = CH_3CHOOB + C_2H_5CHO$ |
| | | $CPENT2ENE + O_3 = C_2H_5CHOOB + CH_3CHO$ |
| Hexene | $C_6H_{12}$ | $HEX1ENE + O_3 = C_4H_9CHO + CH_2OOB$ |
| | | $HEX1ENE + O_3 = HCHO + NC_4H_9CHOOA$ |

**Table R3** Production reactions of stabilized Criegee intermediate (sCI) and OH radical

| Species | Production Reactions of sCI | Production Reactions OH radical |
|---|---|---|
| $CH_2OOB$ | $CH_2OOB = CH_2OO$ | $CH_2OOB = HO_2 + CO + OH$ |
| $CH_2OOD$ | $CH_2OOD = CH_2OO$ | $CH_2OOD = HO_2 + CO + OH$ |
| $CH_2OOE$ | $CH_2OOE = CH_2OO$ | $CH_2OOE = HO_2 + CO + OH$ |
| $ACROOA$ | $ACROOA = ACROO$ | $ACROOA = HO_2 + CO + HCHO + CO + OH$ |
| $MVKOOA$ | $MVKOOA = MVKOO$ | $MVKOOA = OH + MVKO_2$ |
| $MACROOA$ | $MACROOA = MACROO$ | $MACROOA = OH + CO + CH_3CO_3 + HCHO$ |
| $CH_3CHOOA$ | $CH_3CHOOA = CH_3CHOO$ | $CH_3CHOOA = CH_3O_2 + CO + OH$ |
| $CH_3CHOOB$ | $CH_3CHOOB = CH_3CHOO$ | $CH_3CHOOB = CH_3O_2 + CO + OH$ |
| $C_2H_5CHOOA$ | $C_2H_5CHOOA = C_2H_5CHOO$ | $C_2H_5CHOOA = C_2H_5O_2 + CO + OH$ |
| $C_2H_5CHOOB$ | $C_2H_5CHOOB = C_2H_5CHOO$ | $C_2H_5CHOOB = C_2H_5O_2 + CO + OH$ |

| | | |
|---|---|---|
| $C_3H_7CHOOA$ | $C_3H_7CHOOA = C_3H_7CHOO$ | $C_3H_7CHOOA = NC_3H_7O_2 + CO + OH$ |
| $NC_4H_9CHOOA$ | $NC_4H_9CHOOA = NC_4H_9CHOO$ | $NC_4H_9CHOOA = NC_4H_9O_2 + CO + OH$ |

4) It is still not clear to me how the authors retrieved the equation in Line 218 (main text) – I would like the authors to show the derivation of this equation.

$$k_{app}[Alkene][O_3][SO_2]=[SA]CS+\beta[SA]^2 \quad \text{(Line 218 in the main text)}$$

Response: Since the detailed sources of SA in the real atmosphere is really complex, e.g., the varied role of each alkene in producing OH radical, sCI, and SA, we did not intend to precisely calculate the $P_{SA}$ from each of these individual sources. But instead, we use an approximation as indicated by the equation. As mentioned in the manuscript, the two terms on the right side denote the sink of SA, i.e., the condensation of SA onto particles and the collision of SA monomers with each other to form SA dimers. The term on the left side is an empirical one that accounts for the formation from the ozonolysis-of-alkenes-initiated oxidation of $SO_2$. Here, both the OH radical produced from the decomposition of Criegee intermediate (CI) (R3) and stabilized Criegee intermediate (sCI) (R2) are potentially important for oxidizing $SO_2$. As such oxidation is very complicated and reaction dynamics are largely unknown, precise calculation of different reactions is impossible. Therefore, we apply an empirical parameter, i.e., apparent reaction constant ($k_{app}$) to show the general rate of all reactions. The derivation of $k_{app}$ is shown below.

The non-photochemical oxidation pathway for $SO_2$ to form SA mainly includes the following reactions (which have been listed in the manuscript, Line 75, 77, 80-83, Page 3):

$$Alkene+O_3 \xrightarrow{k_1} \varphi sCI+(1-\varphi)CI+RCHO \qquad R1$$

$$sCI+SO_2 \xrightarrow{k_2} SO_3+RCHO \qquad R2$$

$$CI \xrightarrow{k_3} OH+R_1COR_2 \qquad R3$$

$$SO_2+OH \xrightarrow{k_4} HSO_3 \qquad R4$$

$$HSO_3+O_2 \xrightarrow{k_5} SO_3+HO_2 \qquad R5$$

$$SO_3+2H_2O \xrightarrow{k_6} H_2SO_4+H_2O \qquad R6$$

where $k_i$ is the rate constant of each reaction, $\varphi$ is the yield of sCI in the ozonolysis of alkenes, and CI is the chemically activated Criegee intermediate. Besides, there are four additional reactions concerning the loss of sCI, CI and OH:

$$sCI+X \xrightarrow{k_{XX}} \ldots \qquad R7$$

$$sCI \xrightarrow{k_X} \ldots \qquad R8$$

$$CI \xrightarrow{k_Y} \ldots \qquad R9$$

$$Z+OH \xrightarrow{k_{ZZ}} \ldots \qquad R10$$

where $k_{XX}$ is the bimolecular reaction constant for sCI with other species except from $SO_2$, $k_X$ is the decomposition rate of sCI, $k_Y$ is the deposition rate of CI which do not generate OH, and $k_{ZZ}$ is the bimolecular constant reaction for OH with other species excluding $SO_2$.

Then the net production rate of SA can be expressed as:

$$P_{SA}=k_6[SO_3][H_2O]^2 \tag{1}$$

sCI, CI, OH, $SO_3$ and $HSO_3$ are short-lived intermediates, so that a PSS assumption can be applied. PSS equations for sCI, CI, OH, $SO_3$ and $HSO_3$ are as follows:

$$\text{sCI:} \quad k_1[\text{Alkene}][O_3]\varphi=k_2[\text{sCI}][SO_2]+k_{XX}[\text{sCI}][X]+k_X[\text{sCI}] \tag{2}$$

$$\text{CI:} \quad k_1[\text{Alkene}][O_3](1-\varphi)=k_3[\text{CI}]+k_Y[\text{CI}] \tag{3}$$

$$\text{OH:} \quad k_3[\text{CI}]=k_4[SO_2][\text{OH}]+k_{ZZ}[Z][\text{OH}] \tag{4}$$

$$\text{SO}_3: \quad k_2[\text{sCI}][SO_2]+k_5[HSO_3][O_2]=k_6[SO_3][H_2O]^2 \tag{5}$$

$$\text{HSO}_3: \quad k_4[SO_2][\text{OH}]=k_5[HSO_3][O_2] \tag{6}$$

then, the concentrations of sCI, CI and OH are calculated as:

$$[\text{sCI}]=\frac{k_1[\text{Alkene}][O_3]\varphi}{k_2[SO_2]+k_{XX}[X]+k_X} \tag{7}$$

$$[\text{CI}]=\frac{k_1[\text{Alkene}][O_3](1-\varphi)}{k_3+k_Y} \tag{8}$$

$$[\text{OH}]=\frac{k_3[\text{CI}]}{k_4[SO_2]+k_{ZZ}[Z]} \tag{9}$$

bring (5), (6), (7), (8) and (9) to (1) gives:

$$\begin{cases} P_{SA} = k_{app}[\text{Alkene}][O_3][SO_2] \\ k_{app} = k_1k_2\varphi\tau_{sCI}+k_1k_4(1-\varphi)\tau_{OH}\dfrac{k_3}{k_3+k_Y} \\ \tau_{sCI} = \dfrac{1}{k_2[SO_2]+k_{XX}[X]+k_X} \\ \tau_{OH} = \dfrac{1}{k_4[SO_2]+k_{ZZ}[Z]} \end{cases} \tag{10}$$

The first term and second term for $k_{app}$ in equation (10) represent the sCI and non-photochemical OH oxidation pathways respectively. $k_1$, $k_2$ and $k_4$ are chosen to be $1.29\times10^{-17}$ cm$^3$ s$^{-1}$, $8.0\times10^{-13}$ cm$^3$ s$^{-1}$, and $1.38\times10^{-12}$ cm$^3$ s$^{-1}$ separately according to previous studies (Sipila et al., 2014;Mauldin et al., 2012;Wine et al., 1984). $\dfrac{k_3}{k_3+k_Y}$ is the fraction of CI decomposition leading the formation of OH radical, which is around 0.4 according to MCMv3.3.1. $\tau_{sCI}$ and $\tau_{OH}$ are the lifetimes of sCI and OH radical respectively.

It can be seen that $k_{app}$ is highly environment-dependent, as $k_{app}$ is not only influenced by the yields of OH radical and sCI, but also determined by the rate constants of various reactions as well as the concentration of enormous atmospheric species. As discussed in the response to Comment #3, these parameters for each individual alkene are with large uncertainty, and hence, this "bulk oxidant" method is an appropriate option. It should be noted that, the 25%, median, and 75% values of $\tau_{sCI}$ during nighttime under clean condition were 0.012 s, 0.017 s, and 0.021 s, respectively. Similarly, the 25%, median, and 75% values of $\tau_{OH}$ were $1.27\times10^{-3}$ s, $1.73\times10^{-3}$ s and $2.42\times10^{-3}$ s, respectively. The variation of $\tau_{sCI}$ and $\tau_{OH}$ (roughly $\pm$ 50%) could be even smaller than the variation of the rate constants for different alkenes. So, the value of $k_{app}$ may depend more on the mixing ratios of different alkenes.

Besides, the rate constants concerning the lifetime calculation for reactions of sCI with $SO_2$, $H_2O$, CO, $NO_2$, and NO as well as OH radical with VOCs, $O_3$, $SO_2$, CO, $NO_2$ and NO are listed below in Table S4.

**Table R4** Rate constants for reactions of sCI and OH radical used for lifetime calculation

| Compound | Reaction Constant ($cm^{-3}$ $s^{-1}$)* | | Reference |
|---|---|---|---|
| Propylene | $k_{OH}$ | 3.19E-11 | |
| Butadiene | $k_{OH}$ | 7.45E-11 | |
| 1-Butene | $k_{OH}$ | 3.53E-11 | |
| Trans-2-Butene | $k_{OH}$ | 7.35E-11 | |
| Cis-2-Butene | $k_{OH}$ | 6.37E-11 | |
| Isoprene | $k_{OH}$ | 1.11E-10 | |
| 1-Pentene | $k_{OH}$ | 3.56E-11 | |
| Trans-2-Pentene | $k_{OH}$ | 6.69E-11 | |
| Cis-2-Pentene | $k_{OH}$ | 6.54E-11 | |
| 1-Hexene | $k_{OH}$ | 3.70E-11 | |
| Butane | $k_{OH}$ | 2.19E-12 | |
| Acetone | $k_{OH}$ | 1.80E-13 | |
| Pentane | $k_{OH}$ | 3.63E-12 | |
| Methyl Ethyl Ketone (MEK) | $k_{OH}$ | 1.11E-12 | |
| Hexane | $k_{OH}$ | 1.77E-12 | |
| Octane | $k_{OH}$ | 8.30E-12 | |
| Nonane | $k_{OH}$ | 9.68E-12 | |
| Decane | $k_{OH}$ | 1.08E-11 | |
| Undecane | $k_{OH}$ | 1.29E-11 | |
| Dodecane | $k_{OH}$ | 1.39E-11 | MCMv3.3.1 |
| Benzene | $k_{OH}$ | 1.16E-12 | |
| Toluene | $k_{OH}$ | 6.13E-12 | |
| Styrene | $k_{OH}$ | 5.80E-11 | |
| m-Xylene | $k_{OH}$ | 2.31E-11 | |
| p-Xylene | $k_{OH}$ | 1.43E-11 | |
| o-Xylene | $k_{OH}$ | 1.36E-11 | |
| Ethylbenzene | $k_{OH}$ | 7.00E-12 | |
| 1,3,5-Trimethylbenzene | $k_{OH}$ | 5.67E-11 | |
| 1,2,4-Trimethylbenzene | $k_{OH}$ | 3.25E-11 | |
| 1,2,3-Trimethylbenzene | $k_{OH}$ | 3.27E-11 | |
| m-Ethyltoluene | $k_{OH}$ | 1.86E-11 | |
| p-Ethyltoluene | $k_{OH}$ | 1.18E-11 | |
| o-Ethyltoluene | $k_{OH}$ | 1.19E-11 | |
| Propylbenzene | $k_{OH}$ | 6.30E-12 | |
| Heptane | $k_{OH}$ | 6.86E-12 | |
| Chloroform | $k_{OH}$ | 8.71E-14 | |
| $O_3$ | $k_{OH}$ | 5.75E-14 | |
| $SO_2$ | $k_{OH}$ | 1.37E-12 | |
| CO | $k_{OH}$ | 5.64E-11 | |

| | | | |
|---|---|---|---|
| $NO_2$ | $k_{OH}$ | 3.65E-11 | |
| NO | $k_{OH}$ | 1.81E-10 | |
| $SO_2$ | $k_{sCI}$ | 8.00E-13 | (Mauldin et al., 2012) |
| $H_2O$ | $k_{sCI}$ | 1.20E-15 | (Newland et al., 2015) |
| CO | $k_{sCI}$ | 1.20E-15 | MCMv3.3.1 |
| $NO_2$ | $k_{sCI}$ | 1.00E-15 | MCMv3.3.1 |
| NO | $k_{sCI}$ | 1.00E-14 | MCMv3.3.1 |
| sCI | $k_{sCI-dec}$ | 12.0 s | (Newland et al., 2015) |

* The unit for rate constant is $cm^{-3}$ $s^{-1}$ if not specified.

**References**

Ahrens, J., Carlsson, P. T., Hertl, N., Olzmann, M., Pfeifle, M., Wolf, J. L., and Zeuch, T.: Infrared detection of Criegee intermediates formed during the ozonolysis of beta-pinene and their reactivity towards sulfur dioxide, Angew Chem Int Ed Engl, 53, 715-719, 10.1002/anie.201307327, 2014.

Almeida, J., Schobesberger, S., Kuerten, A., Ortega, I. K., Kupiainen-Maatta, O., Praplan, A. P., Adamov, A., Amorim, A., Bianchi, F., Breitenlechner, M., David, A., Dommen, J., Donahue, N. M., Downard, A., Dunne, E., Duplissy, J., Ehrhart, S., Flagan, R. C., Franchin, A., Guida, R., Hakala, J., Hansel, A., Heinritzi, M., Henschel, H., Jokinen, T., Junninen, H., Kajos, M., Kangasluoma, J., Keskinen, H., Kupc, A., Kurten, T., Kvashin, A. N., Laaksonen, A., Lehtipalo, K., Leiminger, M., Leppa, J., Loukonen, V., Makhmutov, V., Mathot, S., McGrath, M. J., Nieminen, T., Olenius, T., Onnela, A., Petaja, T., Riccobono, F., Riipinen, I., Rissanen, M., Rondo, L., Ruuskanen, T., Santos, F. D., Sarnela, N., Schallhart, S., Schnitzhofer, R., Seinfeld, J. H., Simon, M., Sipila, M., Stozhkov, Y., Stratmann, F., Tome, A., Troestl, J., Tsagkogeorgas, G., Vaattovaara, P., Viisanen, Y., Virtanen, A., Vrtala, A., Wagner, P. E., Weingartner, E., Wex, H., Williamson, C., Wimmer, D., Ye, P., Yli-Juuti, T., Carslaw, K. S., Kulmala, M., Curtius, J., Baltensperger, U., Worsnop, D. R., Vehkamaki, H., and Kirkby, J.: Molecular understanding of sulphuric acid-amine particle nucleation in the atmosphere, Nature, 502, 359-+, 10.1038/nature12663, 2013.

Jayne, J. T., Pöschl, U., Chen, Y.-m., Dai, D., Molina, L. T., Worsnop, D. R., Kolb, C. E., and Molina, M. J.: Pressure and Temperature Dependence of the Gas-Phase Reaction of SO3 with H2O and the Heterogeneous Reaction of SO3 with H2O/H2SO4 Surfaces, The Journal of Physical Chemistry A, 101, 10000-10011, 10.1021/jp972549z, 1997.

Kirkby, J., Curtius, J., Almeida, J., Dunne, E., Duplissy, J., Ehrhart, S., Franchin, A., Gagne, S., Ickes, L., Kuerten, A., Kupc, A., Metzger, A., Riccobono, F., Rondo, L., Schobesberger, S., Tsagkogeorgas, G., Wimmer, D., Amorim, A., Bianchi, F., Breitenlechner, M., David, A., Dommen, J., Downard, A., Ehn, M., Flagan, R. C., Haider, S., Hansel, A., Hauser, D., Jud, W., Junninen, H., Kreissl, F., Kvashin, A., Laaksonen, A., Lehtipalo, K., Lima, J., Lovejoy, E. R., Makhmutov, V., Mathot, S., Mikkila, J., Minginette, P., Mogo, S., Nieminen, T., Onnela, A., Pereira, P., Petaja, T., Schnitzhofer, R., Seinfeld, J. H., Sipila, M., Stozhkov, Y., Stratmann, F., Tome, A., Vanhanen, J., Viisanen, Y., Vrtala, A., Wagner, P. E., Walther, H., Weingartner, E., Wex, H., Winkler, P. M., Carslaw, K. S., Worsnop, D. R., Baltensperger, U., and Kulmala, M.: Role of sulphuric acid, ammonia and galactic cosmic rays in atmospheric aerosol nucleation, Nature, 476, 429-U477, 10.1038/nature10343, 2011.

Mauldin, R. L., III, Berndt, T., Sipilae, M., Paasonen, P., Petaja, T., Kim, S., Kurten, T., Stratmann, F., Kerminen, V. M., and Kulmala, M.: A new atmospherically relevant oxidant of sulphur dioxide, Nature, 488, 193-196, 10.1038/nature11278, 2012.

Newland, M. J., Rickard, A. R., Vereecken, L., Munoz, A., Rodenas, M., and Bloss, W. J.: Atmospheric isoprene ozonolysis: impacts of stabilised Criegee intermediate reactions with SO2, H2O and dimethyl sulfide, Atmospheric Chemistry and Physics, 15, 9521-9536, 10.5194/acp-15-9521-2015, 2015.

Sipila, M., Jokinen, T., Berndt, T., Richters, S., Makkonen, R., Donahue, N. M., Mauldin, R. L., III, Kurten, T., Paasonen, P., Sarnela, N., Ehn, M., Junninen, H., Rissanen, M. P., Thornton, J., Stratmann, F., Herrmann, H., Worsnop, D. R., Kulmala, M., Kerminen, V. M., and Petaja, T.: Reactivity of stabilized Criegee intermediates (sCIs) from isoprene and monoterpene ozonolysis toward SO2 and organic acids, Atmospheric Chemistry and Physics, 14, 12143-12153, 10.5194/acp-14-12143-2014, 2014.

Wine, P. H., Thompson, R. J., Ravishankara, A. R., Semmes, D. H., Gump, C. A., Torabi, A., and Nicovich, J. M.: Kinetics of the reaction OH + SO2 + M .fwdarw. HOSO2 + M. Temperature and pressure dependence in the fall-off region, The Journal of Physical Chemistry, 88, 2095-2104, 10.1021/j150654a031, 1984.

Yao, L., Garmash, O., Bianchi, F., Zheng, J., Yan, C., Kontkanen, J., Junninen, H., Mazon, S. B., Ehn, M., Paasonen, P., Sipila, M., Wang, M. Y., Wang, X. K., Xiao, S., Chen, H. F., Lu, Y. Q., Zhang, B. W., Wang, D. F., Fu, Q. Y., Geng, F. H., Li, L., Wang, H. L., Qiao, L. P., Yang, X., Chen, J. M., Kerminen, V. M., Petaja, T., Worsnop, D. R., Kulmala, M., and Wang, L.: Atmospheric new particle formation from sulfuric acid and amines in a Chinese megacity, Science, 361, 278-+, 10.1126/science.aao4839, 2018.

Yao, L., Fan, X., Yan, C., Kurten, T., Daellenbach, K. R., Li, C., Wang, Y., Guo, Y., Dada, L., Rissanen, M. P., Cai, J., Tham, Y. J., Zha, Q., Zhang, S., Du, W., Yu, M., Zheng, F., Zhou, Y., Kontkanen, J., Chan, T., Shen, J., Kujansuu, J. T., Kangasluoma, J., Jiang, J., Wang, L., Worsnop, D. R., Petaja, T., Kerminen, V. M., Liu, Y., Chu, B., He, H., Kulmala, M., and Bianchi, F.: Unprecedented Ambient Sulfur Trioxide (SO3) Detection: Possible Formation Mechanism and Atmospheric Implications, Environ Sci Technol Lett, 7, 809-818, 10.1021/acs.estlett.0c00615, 2020.